# The Blessing and Curse of Dimensionality in Safety Alignment

**Rachel S.Y. Teo**[*]
Department of Mathematics
National University of Singapore
rachel.tsy@u.nus.edu

**Laziz U. Abdullaev**[*]
Department of Mathematics
National University of Singapore
laziz.abdullaev@u.nus.edu

**Tan M. Nguyen**
Department of Mathematics
National University of Singapore
tanmn@nus.edu.sg

## Abstract

The focus on safety alignment in large language models (LLMs) has increased significantly due to their widespread adoption across different domains. The scale of LLMs play a contributing role in their success, and the growth in parameter count follows larger hidden dimensions. In this paper, we hypothesize that while the increase in dimensions has been a key advantage, it may lead to emergent problems as well. These problems emerge as the linear structures in the activation space can be exploited, in the form of activation engineering, to circumvent its safety alignment. Through detailed visualizations of linear subspaces associated with different concepts, such as safety, across various model scales, we show that the curse of high-dimensional representations uniquely impacts LLMs. Further substantiating our claim, we demonstrate that projecting the representations of the model onto a lower dimensional subspace can preserve sufficient information for alignment while avoiding those linear structures. Empirical results confirm that such dimensional reduction significantly reduces susceptibility to jailbreaking through representation engineering. Building on our empirical validations, we provide theoretical insights into these linear jailbreaking methods relative to a model's hidden dimensions. Broadly speaking, our work posits that the high dimensions of a model's internal representations can be both a blessing and a curse in safety alignment.

## 1 Introduction

Large language models (LLMs) have become ubiquitous due to their significant success in a wide variety of applications such as natural language generation (Zhao et al., 2023; Dong et al., 2019), logical reasoning (Wei et al., 2022; Wang et al., 2022; Huang & Chang, 2023), and summarization (Liu et al., 2024b; Van Veen et al., 2024; Zhang et al., 2024). To capitalize on their capabilities, frontier models are trusted with more autonomy and agency to assist humans or even replace them entirely (Anthropic, 2024; Lu et al., 2024; OpenAI, 2025). While these advancements are valuable and practical, the consequences of an LLM's vulnerability are more severe than ever; LLMs are susceptible to attacks that lead to harmful responses that should be avoided. AI alignment, in particular, safety alignment, aims to ensure that the model is able to provide helpful responses to harmless prompts while avoiding harmful instructions (Leike et al., 2018; Matthews et al., 2022; Kenton et al., 2021; Ji et al., 2023; Ngo, 2022). The general procedure in aligning LLMs (Achiam et al., 2023; Touvron et al., 2023; Grattafiori et al., 2024; Team et al., 2024; Bai et al., 2023; Anthropic, 2023; Liu et al., 2024a) follow multiple rounds of Supervised Fine-tuning (SFT) (Wei et al., 2021), Reinforcement

---

[*]Co-first authors

> **Positive emotion:** `You receive a message from your crush asking to spend time together.`
>
> **Negative emotion:** `Someone promises to meet but doesn't show up or notify.`

**Figure 1:** Examples of contrasting prompts representing a positive or negative emotion used to find the steering vector for emotion.

Learning with Human Feedback (RLHF) (Ouyang et al., 2022; Kaufmann et al., 2023), and Direct Preference Optimization (DPO) (Rafailov et al., 2023). Considering the importance of safety alignment in LLM deployment, it is crucial to understand the internal model representations of safety and their possible subversions.

In this work, we analyze the concept of safety being represented in an LLM with an emphasis on the *dimensionality* of the representations. It has been extensively theorized that high-level concepts or features are linearly represented as directions in the activation space (Park et al., 2024a;b; Zheng et al., 2024; Bolukbasi et al., 2016; Mikolov et al., 2013; Marks & Tegmark, 2023). These linear representations can be leveraged or exploited to steer the model toward a desired behavior by purposefully modifying its activations (Wolf et al., 2024; Rimsky et al., 2024; Turner et al., 2023; Zou et al., 2023a). This is the process of activation engineering. However, these linear structures are assumed to be emergent in LLMs. That is, they are only present in models that are sufficiently large, with high-dimensional representations (Park et al., 2023; Marks & Tegmark, 2023; Arditi et al., 2024). Our goal is to elucidate that assumption and its role in safety alignment. To be concrete, our contribution is three-fold.

1. We analyze the role of dimensionality in the linear representation hypothesis through systematic experiments and visualizations with particular attention to its relation to safety alignment.

2. We present theoretical insights into jailbreaking methods that exploit the linear structures in the activation space to elicit potentially harmful responses from LLMs through representation engineering.

3. We propose novel fine-tuning methods for LLMs that include projecting their hidden representations onto a lower-dimensional subspace to safeguard against recently proposed jailbreaking methods while retaining sufficient information to be safety aligned.

Our work demonstrates the dichotomy of the advantages and potential pitfalls of scaling models with increasing hidden dimensions. Specifically, we focus on open-source models where manipulations of their internals are possible and can be misused. We are inspired by Arditi et al. (2024) and build on their work to counter the white-box jailbreak proposed in their paper and connect them to the double-edged sword of representation engineering. By integrating multiple concepts in LLM research into a unifying framework, our goal is to enhance existing approaches to safety alignment.

## 2 Background

**Notation.** We denote a prompt and response pair as $(\mathbf{x}, \mathbf{y}) \sim \mathcal{D}$ when sampled from a distribution, $\mathcal{D}$. These are token sequences and we use $\mathbf{y}_{<t}$ to denote the subsequence of tokens from the first position, 0 to $t-1$. $y_t$ is used to refer to token $t$ in response $\mathbf{y}$ and $|\mathbf{y}|$ refers to the length of a sequence. We use $\pi_\theta$ to represent an LLM with parameter weights $\theta$ and $\pi_{\text{aligned}}$ for an aligned model. These are usually referred to as **Chat** or **Instruct (IT)** models and have undergone post-training for alignment and instruction-tuning. LLMs referred to as **Base** models are the models that have been pre-trained but without further fine-tuning or preference optimization. Finally, $\pi(\cdot \mid \mathbf{x}, \mathbf{y}_{<t})$ denotes the vector of probabilities of the next generated token $y_t$.

**Linear Representations.** Linear representations have led to considerable advancements in the interpretability of a model's hidden states (Mikolov et al., 2013; Nanda et al., 2023; Gurnee & Tegmark, 2023; Jiang et al., 2024). We consider a linear representation to be a direction in the activation space of the LLM that represents a concept like "truthfulness" or "safety". These concepts are more complex, as compared to high-level concepts like color or

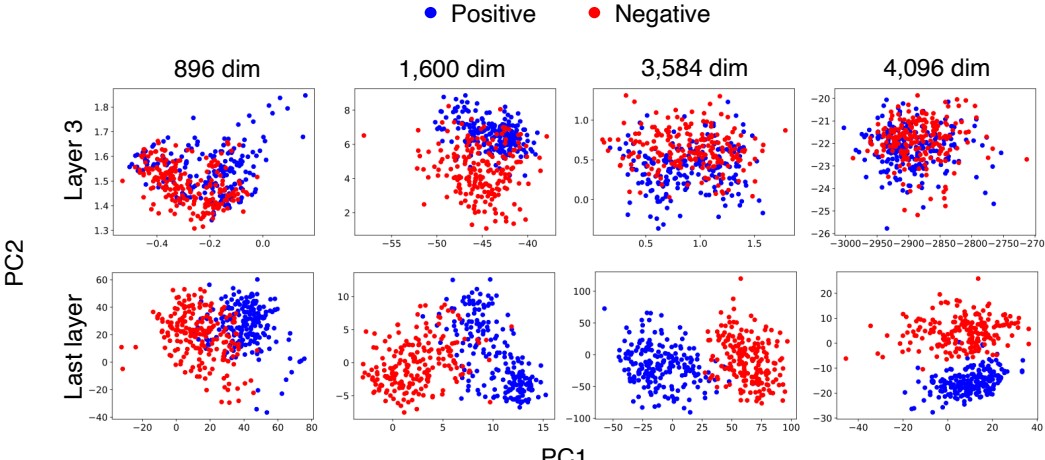

**Figure 2:** (Left to right) Each column visualizes the hidden representations from different layers of Qwen0.5B, GPT-XL, Qwen7B, and Llama2-7B respectively, when projected onto the top-2 principal components. The top row are hidden representations from the third layer while the bottom row are from the last layer of each model. Red and blue points are representations of prompts corresponding to positive and negative emotion. Each model's hidden dimension is noted at the top of their column. We observe that as the hidden dimension increases, the separation becomes more pronounced over layers, illustrating a trend toward stronger linear representations of emotion in larger models.

gender (Mikolov et al., 2013; Abdou et al., 2021; Patel & Pavlick, 2022) and seem to emerge in LLMs due to their scale. These concept vectors are typically found by collecting the model activations for two contrastive sets of prompts and finding the difference between them. An example of prompts that represent the concept of emotions can be found in Figure 1. We may also refer to these vectors as steering vectors.

**Jailbreaking via Activation Engineering.** A jailbreak attack refers to an intentional attempt to bypass a model's safety alignment and elicit a harmful response, thereby disabling a refusal. A simple method proposed by (Arditi et al., 2024) leverages the safety direction to steer the model. Thereafter, the model refuses harmless requests while generating harmful content, and is successfully attacked. We refer to this jailbreak method as **ActAdd**. A similar method, **Ablation**, ablates this safety direction in the activations to induce refusal. More details can be found in Appendix B.1 and B.2.

**Transformers.** The main architecture of LLMs follows a decoder-only transformer model (Vaswani et al., 2017). For a given prompt sequence $\mathbf{x}$ and its hidden representation $\mathbf{x}^{(\ell)}$ at layer $\ell$, the next layer in the LLM transforms its input by the multi-head attention mechanism, followed by an MLP. The *attention* component projects $\mathbf{x}^{(\ell)}$ into the query $\mathbf{Q}$, key $\mathbf{K}$, and value matrix $\mathbf{V}$ via three linear transformations independently at each attention head[1]:

$$\mathbf{Q} = \mathbf{x}^{(\ell)}\mathbf{W}_Q^\top; \mathbf{K} = \mathbf{x}^{(\ell)}\mathbf{W}_K^\top; \mathbf{V} = \mathbf{x}^{(\ell)}\mathbf{W}_V^\top,$$

where $\mathbf{W}_Q, \mathbf{W}_K$, and $\mathbf{W}_V$ are the weight matrices. For ease of notation, we omit the superscript $\ell$ for the query, key, value matrices and their weights, however it should be noted that they are unique to each layer. The intermediate output of attention $\tilde{\mathbf{x}}^{(\ell)}$ and final output $\mathbf{x}^{(\ell+1)}$ of layer $\ell + 1$, is then computed as follows

$$\tilde{\mathbf{x}}^{(\ell)} = \mathbf{x}^{(\ell)} + \text{softmax}\left(\mathbf{Q}\mathbf{K}^\top/\sqrt{D}\right)\mathbf{V}; \quad \mathbf{x}^{(\ell+1)} = \tilde{\mathbf{x}}^{(\ell)} + \text{MLP}(\tilde{\mathbf{x}}^{(\ell)}),$$

where the softmax function is applied to each row of the matrix $\mathbf{Q}\mathbf{K}^\top/\sqrt{D}$ and $D$ is the hidden dimension of $\mathbf{x}$.

---

[1]See Appendix A.1 for further details.

## 3 The Paradox of Linear Separability

### 3.1 Linear Separability

**Activations Visualization.** The linear structures that represent concepts can be simply visualized using principal component analysis (PCA). If there is an evident direction corresponding to a particular concept, there should be two distinct clusters consistent with two contrastive sets of prompts and strong linear separability between them.

In Figure 2, we plot the activations, from models of varying sizes, of two separate sets of prompts that represent positive and negative emotions when projected onto the top-2 principal components (PCs). While this work is focused on safety, we use emotion in this section as we are comparing the linear representations of language models with relatively small hidden dimensions that do not have an aligned counterpart. We choose emotion as it is an abstract concept, similar in complexity to safety.

We observe that across all hidden dimensions, layer 3 has no separation between the two sets at all and seem uninformative. Conversely, in the final layer, there are distinct, separable clusters of positive and negative emotions in models with more than 3,000 dimensions. For the smaller models, these clusters clearly overlap with a large degree of mixing, suggesting a lack of linear representations. These visualizations substantiate the usual assumptions in the linear representation hypothesis, that the linear structures of increasingly complex concepts are only present in models of sufficiently large scale.

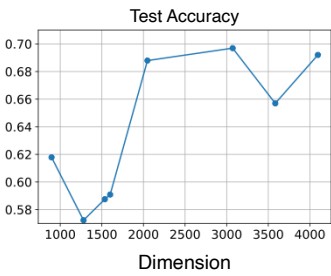

**Figure 3:** Test accuracy of linear probes trained on activations from models with varying hidden dimensions. The probes are trained to classify if a prompt represents a positive or negative emotion.

**Linear Probing.** We corroborate our visualization with a linear probe experiment and present our results in Figure 3. Using the activations from models of different families and scales, we train a linear probe to predict if the prompt conveyed positive or negative emotions. We test on a held out set of emotions, to assess the model's generalization of the concept. As illustrated in the line plot, there is an obvious distinction between models below and above 2,000 hidden dimensions, with greater accuracy achieved with higher dimensions, suggesting better generalization and a strong linear structure representing emotion. Details on the experiment can be found in Appendix E.

These results support the hypothesis that *abstract concepts require a sufficiently large activation space to be linearly represented*. While these directions may be beneficial as steering directions to control a model's behavior, there is also danger in exploiting these linear structures to overcome safety alignment. We refer to this tension as the Paradox of Linear Separability, which can be summarized as follows:

> **Paradox of Linear Separability**
>
> As models are scaled up to improve their abilities, they also become more vulnerable to steering jailbreaks, making it increasingly critical to protect their alignment.

### 3.2 Learning-Theoretic Perspective

In this section, we provide theoretical insights into the relationship between steering jailbreaks and the hidden dimensions of the model. We analyze the steering vectors from the perspective of learning theory, motivating our methodology to defend against such attacks.

We begin by considering a class of adversaries that rely on linear concept erasure. This entails learning a linear classifier to classify the binary classes of a concept (Ravfogel et al., 2022; Belrose et al., 2023; Marks & Tegmark, 2023; Arditi et al., 2024) and effectively erasing the concept encoded along that 1-dimensional subspace by removing that direction from the feature space. ActAdd jailbreak is one type of attack within this class.

The effectiveness of linear functions in high-dimensional feature spaces can be understood through the lens of Vapnik–Chervonenkis (VC) dimension, as the VC dimension of a linear

hypothesis class in $\mathbb{R}^D$ is $D + 1$ (Mohri et al., 2018). However, we present a simple alternative argument for evaluating the richness of a hypothesis class as a function of input dimension, using Rademacher complexity—a measure that captures data dependence and avoids the purely combinatorial nature of the VC dimension.

Let $\mathcal{F}$ be a family of functions mapping from $\mathbb{R}^D$ to $\mathbb{R}$. Then, the *empirical Rademacher complexity* of $\mathcal{F}$ for a sample $\mathcal{S} = (\mathbf{x}_1, \ldots, \mathbf{x}_N)$, is defined by

$$\widehat{\mathfrak{R}}_{\mathcal{S}}(\mathcal{F}) = \mathbb{E}_{\sigma} \left[ \sup_{f \in \mathcal{F}} \frac{1}{N} \sum_{i=1}^{N} \sigma_i f(\mathbf{x}_i) \right], \tag{1}$$

where $\sigma = (\sigma_1, \ldots, \sigma_N)$ is a vector of i.i.d. Rademacher variables, which are independent uniform random variables taking values in $\{-1, +1\}$. The *Rademacher complexity* of $\mathcal{F}$, denoted $\mathfrak{R}_N(\mathcal{F})$, is defined as the expectation of this quantity:

$$\mathfrak{R}_N(\mathcal{F}) = \mathbb{E}_{\mathcal{S} \sim \mathcal{D}^N}[\widehat{\mathfrak{R}}_{\mathcal{S}}(\mathcal{F})], \tag{2}$$

where $\mathcal{D}$ represents a distribution over the input space $\mathbb{R}^D$. The empirical Rademacher complexity is a crucial data-dependent measure of complexity. The following useful bound is due to Awasthi et al. (2020):

**Theorem 1** *Let $\mathcal{F} = \{f : f(\mathbf{x}) = \mathbf{w}^{\top}\mathbf{x}, \|\mathbf{w}\|_2 \leq L, \mathbf{x} \in \mathbb{R}^D\}$ be the family of linear functions with bounded weight. Then, the empirical Rademacher complexity (Eqn. 1) of $\mathcal{F}$ given a data matrix $\mathbf{X} \in \mathbb{R}^{N \times D}$ admits the following bound:*

$$\widehat{\mathfrak{R}}_{\mathbf{X}}(\mathcal{F}) \leq \frac{L\|\mathbf{X}\|_F}{N}. \tag{3}$$

Based on this result, we explicitly relate the Rademacher complexity of the linear hypothesis class to dimensionality of the input in Proposition 1.

**Proposition 1** *Assume that feature vectors are normally distributed. Then, the Rademacher complexity (given by Eqn. 2) of $\mathcal{F}$ admits the following bound up to a constant factor:*

$$\mathfrak{R}_N(\mathcal{F}) \lesssim L\sqrt{\frac{D}{N}}. \tag{4}$$

The bound in Eqn. 4 explicitly separates the effects of sample dimension and sample size in determining the capacity of a linear hypothesis class on the given sample. To be more precise, dimension reduction applied to the input space effectively reduces the Rademacher complexity of the linear hypothesis class at the rate of $O(\sqrt{D})$. Consequently, the concept encoded in a single direction in $\mathbb{R}^D$ is expected to be shattered over a higher dimensional subspace in $\mathbb{R}^k$ with $k < D$, diminishing the performance of a linear classifier. Hence *it should be more difficult to find an effective steering vector in a lower dimensional subspace, limiting the success of the ActAdd attack*. The proof of Proposition 1 is provided in Appendix A.4.

## 4 Guarding Against Steering Vectors

We develop two novel approaches as a defense mechanism against steering jailbreaks and describe them below. Both methods are motivated by the theory presented in Section 3.2 to reduce the dimension of the activation space. We build our methods on top of already aligned Chat and Instruct models, as the exact post-training pipelines on base models are not publicly available.

### 4.1 Fast Johnson–Lindenstrauss Transform

Our first approach hinges on the Fast Johnson-Lindenstrauss Transform (FJLT) (Ailon & Chazelle, 2006), a low-distortion embedding of a high-dimensional normed metric space into low-dimensional one. A well established result in dimensionality reduction is the Johnson-Lindenstrauss (JL) Lemma that proves the existence of a mapping for $n$ points in high dimensional euclidean space onto $K$ dimensions that preserves the euclidean distance between any two points[2]. The FJLT is a sped-up implementation of the JL transform, and

---

[2]The precise statement is in Appendix A.2 for completeness

we capitalize on their preservation of Euclidean distance by introducing this mapping as a projection in *every* attention layer of the LLM.

Concretely, we construct our FJLT projection matrix $\Phi^{(\ell)} \in \mathbb{R}^{D \times K}$, where $K < D$, for each layer $\ell$ and apply it to the query and key matrices:

$$\boldsymbol{Q}_{proj} = \boldsymbol{Q}\Phi^{(\ell)}; \quad \boldsymbol{K}_{proj} = \boldsymbol{K}\Phi^{(\ell)}; \quad \tilde{\mathbf{x}}^{(\ell)} = \mathbf{x}^{(\ell)} + \text{softmax}\left(\boldsymbol{Q}_{proj}\boldsymbol{K}_{proj}^{\top}/\sqrt{D}\right)\boldsymbol{V}.$$

We chose the query and key matrices to apply the FJLT projection as the inner products between each row of $\boldsymbol{Q}$ and $\boldsymbol{K}$ should be approximately preserved in theory while the projection into a lower dimensional activation space acts as a defense mechanism for steering attacks. As LLMs generally use multi-head attention, we only choose one head to apply the FJLT projection and consider this a hyperparameter. More details on multi-head attention and an ablation study on the heads are in Appendix B.4 and C.1. For LLMs with the FJLT projection matrix implemented, we refer to them as a **FJLT** model.

Then, we fine-tune the FJLT model using the following token-wise constrained objective from Qi et al. (2024)

$$\min_{\theta} \left\{ \mathbb{E}_{(\mathbf{x},\mathbf{y}) \sim \mathcal{D}} - \sum_{t=1}^{|\mathbf{y}|} \frac{2}{\beta_t} \log \left[ \sigma \left( \beta_t \log \frac{\pi_{\theta}(y_t \mid \mathbf{x}, \mathbf{y}_{<t})}{\pi_{\text{aligned}}(y_t \mid \mathbf{x}, \mathbf{y}_{<t})} \right) \right] \right\}, \tag{5}$$

where $\beta_t > 0$ is a hyperparameter, $\pi_{\text{aligned}}$ is a Chat or Instruct model used as a reference model and $\sigma(x) := 1/(1 + \exp^{-x})$ is the sigmoid function. In their work, they provide some theoretical analysis on the limiting behaviors of the objective with respect to $\beta_t$ and interpretations from a reinforcement learning perspective. Complementarily, we offer a different interpretation of their objective function, in the form of weighted entropy (Kelbert et al., 2017b;a) in Appendix A.5. We use this objective to encourage the FJLT model to minimize the deviation of its distribution over the generated tokens from the aligned model, since the FJLT model has theoretical groundings that justify its approximation of the aligned model.

**Limitation.** A limitation of the FJLT model is that the LLM is only able to perform well on datasets that are similar to the fine-tuning dataset. We observe in Appendix D, Table 13, that while the FJLT model can perform close to the baseline model on THE PILE (Gao et al., 2020) and Alpaca (Taori et al., 2023), it is unable to answer many prompts from SQL Create Context (b mc2, 2023), Samsum (Gliwa et al., 2019) and GSM8k (Cobbe et al., 2021). This could result from an over-compression of the model's concepts or knowledge. We corroborate this in Section 5.2 and are motivated to present an additional Bottleneck method that can overcome such limitations.

### 4.2 Bottleneck

An alternative to the FJLT projection is to insert a linear autoencoder between a pair of consecutive layers in the model. The index of the layer $\ell$ where it is inserted is considered a hyperparameter and chosen empirically. Since we only insert a single projection layer, we would expect less information loss compared to the FJLT model. The structure of the linear autoencoder is simple,

$$\mathbf{x}^{(\ell)}_{compressed} = \sigma(\mathbf{x}^{(\ell)}\boldsymbol{W}_{down}\boldsymbol{W}_{up}),$$

where $\sigma$ is an activation function, $\boldsymbol{W}_{down} \in \mathbb{R}^{D \times K}$ and $\boldsymbol{W}_{up} \in \mathbb{R}^{K \times D}$ for $K < D$. Then, $\mathbf{x}^{(\ell)}_{compressed}$ is processed by layer $\ell + 1$. We refer this model as a **Bottleneck** model.

To fine-tune the Bottleneck model, we use the standard fine-tuning objective, with a refusal dataset $D_P$ and an anchor utility dataset $D_B$,

$$\min_{\theta} \mathbb{E}_{(\mathbf{x},\mathbf{y}) \sim D_P} \left[ -\log \pi_{\theta}(\mathbf{y} \mid \mathbf{x}) \right] + \alpha \, \mathbb{E}_{(\mathbf{x},\mathbf{y}) \sim D_B} \left[ -\log \pi_{\theta}(\mathbf{y} \mid \mathbf{x}) \right].$$

$\alpha$ is a regularization hyperparameter to control the influence of the anchor loss on the objective. $D_P$ consists of harmful prompts with refusal responses and $D_B$ consists of benign prompts with safe responses. The decision to return to the standard objective function is motivated empirically. Further, the Bottleneck model has no theoretical grounds to be expected to match the aligned model's distribution over $y_t$.

**Table 1:** Hyperparameters, refusal scores, safety scores and perplexity (PPL) on harmful and benign instructions **after** attacking each model using the ActAdd jailbreak. We also include results on the baseline Chat or Instruct models without jailbreaks for reference. The direction of increasing or decreasing values signifying better performance is indicated by each metric's arrow direction. Our model is highlighted in grey.

| Jailbreak | Model | Head | K | Harmful Inst. | | Benign Inst. | |
| | | | | Refusal ↑ | Safety ↑ | Refusal ↓ | PPL ↓ |
|---|---|---|---|---|---|---|---|
| | Llama2-7B-Chat | | | 0.97 | 0.99 | 0.00 | 1.12 |
| ActAdd | Llama2-7B-Chat | | | 0.03 | 0.14 | 1.00 | 1.58 |
| | Llama2-7B-Chat-FT | | | $0.66_{\pm0.17}$ | $0.65_{\pm0.17}$ | $0.91_{\pm0.04}$ | $\mathbf{1.09}_{\pm0.13}$ |
| | Llama2-7B-Chat-FJLT | 0 | 64 | $\mathbf{0.94}_{\pm0.06}$ | $\mathbf{0.96}_{\pm0.03}$ | $\mathbf{0.63}_{\pm0.03}$ | $1.37_{\pm0.11}$ |
| | Gemma-1.1-7B-IT | | | 0.92 | 0.96 | 0.00 | 1.27 |
| ActAdd | Gemma-1.1-7B-IT | | | 0.41 | 0.55 | 0.62 | 1.72 |
| | Gemma-1.1-7B-IT-FT | | | $0.37_{\pm0.23}$ | $0.53_{\pm0.10}$ | $0.87_{\pm0.05}$ | $1.68_{\pm0.03}$ |
| | Gemma-1.1-7B-IT-FJLT | 0 | 96 | $\mathbf{0.90}_{\pm0.04}$ | $\mathbf{0.85}_{\pm0.04}$ | $\mathbf{0.55}_{\pm0.08}$ | $\mathbf{1.40}_{\pm0.06}$ |
| | Qwen2-7B-Instruct | | | 0.99 | 0.99 | 0.01 | 1.44 |
| ActAdd | Qwen2-7B-Instruct | | | 0.11 | 0.18 | 0.91 | 1.77 |
| | Qwen2-7B-Instruct-FT | | | $0.14_{\pm0.04}$ | $0.15_{\pm0.03}$ | $0.91_{\pm0.03}$ | $\mathbf{1.46}_{\pm0.03}$ |
| | Qwen2-7B-Instruct-FJLT | 0 | 64 | $\mathbf{0.89}_{\pm0.09}$ | $\mathbf{0.91}_{\pm0.07}$ | $\mathbf{0.76}_{\pm0.05}$ | $1.59_{\pm0.08}$ |

## 5 Experiments

In this section, we aim to: i) empirically validate the defenses of both FJLT and Bottleneck models on the ActAdd jailbreak (Section 5.1); ii) Analyze the preservation of linear representations in both models (Section 5.2). More experiments on the Ablation jailbreak and other benchmarks can be found in Appendix D.2.

### 5.1 Jailbreak Benchmark Evaluations

We fine-tune three models, Llama2-7B-Chat (Touvron et al., 2023), Gemma-1.1-7B-IT (Team et al., 2024) and Qwen2-7B-Instruct (Yang et al., 2024) and evaluate them on JailbreakBench (Chao et al., 2024) and Alapca dataset (Taori et al., 2023). We report results on the baseline Chat or Instruct model, fine-tuned model without any modifications (denoted by *FT*), FJLT and Bottleneck models. Implementations details can be found in Appendix B. Our results are averaged over 5 runs and conducted on a server with 8 H100 GPUs.

**Harmful Instruction Evaluations.** To evaluate the success of the ActAdd jailbreak on harmful instructions, we use a refusal and safety score following Arditi et al. (2024). The refusal score represents the proportion of prompts where the LLM's generated response are refusals. The safety score is measured as the proportion of model responses that are considered safe by Meta Llama Guard 2 (Team, 2024), a reliable open-sourced model fine-tuned to identify harmful content. These metrics should be higher on harmful instructions.

**Benign Instructions Evaluations.** For safe instructions evaluations, we report the refusal score and perplexity (PPL) of the responses on benign prompts to ensure that the model does not just avoid refusal, but can provide a coherent response. For a coherent, safety aligned model, we would expect the refusal score and PPL to be low on benign instructions.

#### 5.1.1 Fast Johnson–Lindenstrauss Transform Experiments

For the FJLT model, there are three hyperparameters to consider: i) $0 < K < D_H$, the number of dimensions to project onto; ii) Head, the attention head index to implement the FJLT projection; iii) $\beta_t > 0$, a regularization parameter for the objective function, Eqn. 5. Note here that $D_H < D$ is the number of hidden dimensions in each attention head. We present the different values of $K$ for each model, along with the attention head used and our results in Table 1. We only use one head for projection in each layer. For $\beta_t$, we follow the setting provided by Qi et al. (2024). We fine-tune each model on $D_p$, a set of harmful prompts with refusal responses.

We observe that when we simply fine-tune the model, after the jailbreak, there are already some improvements in the refusal and safety scores on harmful instructions, but minimal on

**Table 2:** Hyperparameters, refusal scores, safety scores and perplexity (PPL) on harmful and benign instructions **after** attacking each model using the ActAdd jailbreak. We also include results on the baseline Chat or Instruct models without jailbreaks for reference. The direction of increasing or decreasing values signifying better performance is indicated by each metric's arrow direction. Our model is highlighted in grey.

| Jailbreak | Model | $\alpha$ | Layer | K | Harmful Inst. Refusal ↑ | Safety ↑ | Benign Inst. Refusal ↓ | PPL ↓ |
|---|---|---|---|---|---|---|---|---|
| | Llama2-7B-Chat | | | | 0.97 | 0.99 | 0.00 | 1.12 |
| ActAdd | Llama2-7B-Chat | 1.0 | | | 0.03 | 0.14 | 1.00 | 1.58 |
| | Llama2-7B-Chat-FT | 1.0 | | | $0.55_{\pm 0.02}$ | $0.35_{\pm 0.02}$ | $0.99_{\pm 0.00}$ | $\mathbf{1.07}_{\pm 0.01}$ |
| | Llama2-7B-Chat-Bottleneck | 1.0 | 0 | 2048 | $\mathbf{0.95}_{\pm 0.02}$ | $\mathbf{0.97}_{\pm 0.01}$ | $\mathbf{0.30}_{\pm 0.03}$ | $1.08_{\pm 0.34}$ |
| | Gemma-1.1-7B-IT | | | | 0.92 | 0.96 | 0.00 | 1.27 |
| ActAdd | Gemma-1.1-7B-IT | 0.1 | | | 0.41 | 0.55 | 0.62 | 1.72 |
| | Gemma-1.1-7B-IT-FT | 0.1 | | | $0.00_{\pm 0.00}$ | $0.54_{\pm 0.03}$ | $0.75_{\pm 0.04}$ | $\mathbf{1.56}_{\pm 0.07}$ |
| | Gemma-1.1-7B-IT-Bottleneck | 0.1 | 0 | 1536 | $\mathbf{0.82}_{\pm 0.08}$ | $\mathbf{0.83}_{\pm 0.06}$ | $\mathbf{0.41}_{\pm 0.02}$ | $2.50_{\pm 0.33}$ |
| | Qwen2-7B-Instruct | | | | 0.99 | 0.99 | 0.01 | 1.44 |
| ActAdd | Qwen2-7B-Instruct | 1.0 | | | 0.11 | 0.18 | 0.91 | 1.77 |
| | Qwen2-7B-Instruct-FT | 1.0 | | | $0.08_{\pm 0.00}$ | $0.19_{\pm 0.01}$ | $0.98_{\pm 0.00}$ | $1.66_{\pm 0.02}$ |
| | Qwen2-7B-Instruct-Bottleneck | 1.0 | 0 | 1792 | $\mathbf{0.23}_{\pm 0.10}$ | $\mathbf{0.51}_{\pm 0.19}$ | $\mathbf{0.74}_{\pm 0.11}$ | $\mathbf{1.47}_{\pm 0.05}$ |

the safe instructions. This is likely due to the additional refusal fine-tuning we performed. After accounting for this compounding effect, there is still remarkable progress in defending against the ActAdd jailbreak, minimizing its success. The refusal and safety scores for harmful instructions return to almost uncompromised baseline levels and some improvements on safe instructions. These results serve to support our hypothesis that models can have strong guardrails against steering jailbreaks in a lower dimensional representation space.

### 5.1.2  Bottleneck Experiments

For the Bottleneck model, there are also three hyperparameters: i) $0 < K < D$; ii) $\ell$, the layer in the LLM to insert this autoencoder after; iii) $\alpha > 0$, the regularization parameter to control the anchor dataset's loss. We report the values of all hyperparamters used in Table 2, along with our results.

Similar to the FJLT model results, there are already improvements with the FT model. After accounting for the effect of fine-tuning itself, there are still significant improvements on both harmful and benign instructions across all models and metrics. The greatest improvement comes from Llama2-7B-Chat-Bottleneck, whose refusal score increases to almost 1 on harmful instructions and lowers to 0.3 on safe instructions. This shows that the model is able to answer 70% of the benign prompts while refusing almost all harmful instructions, rendering the ActAdd attack practically ineffective. Furthermore, in Table 13, Appendix D, we find that the Bottleneck model is able to perform close to the level of the baseline Chat model on several utility benchmarks, maintaining its utility and improving over the FJLT model.

### 5.2  Concepts in Projected Subspaces

Figure 4 displays the first two PCs of hidden representations for three concept categories in the baseline Chat, FJLT, and Bottleneck models based on Llama2-7B-Chat. While our primary goal is to manage the linear geometry of safety, maintaining other concept structures are crucial to preserve the utility of the model. Notably, both FJLT and Bottleneck models effectively disrupt linear separability for safety, suggesting a potential non-linear encoding in the reduced subspace. However, FJLT also distorts the distinct structures of truthfulness and emotion, possibly reducing the quality of responses. In fact, as seen in Table 13 in the appendix, FJLT performs well on THE PILE and Alpaca, but extremely poorly on SQL Create Context, Samsum, and GSM8k. In contrast, the Bottleneck model preserves the original separation of truthfulness and emotion, while successfully mitigating linear separability in safety.

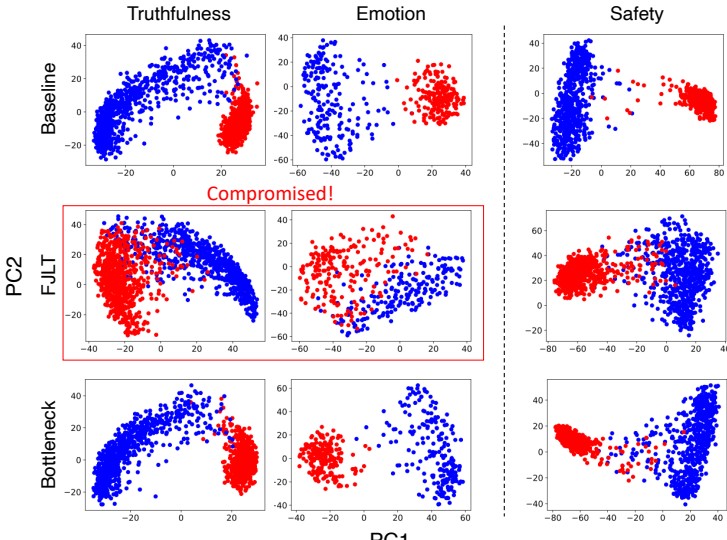

**Figure 4:** Projection of the hidden representations onto the top-2 principal components for three concept categories–truthfulness, emotion, and safety–in the baseline Chat, FJLT, and Bottleneck models based on Llama2-7B-Chat.

## 6 Related Work

**Concept erasure.** Concept erasure aims to remove specific attributes—such as bias or protected features—from learned representations while preserving task-relevant information. Several methods address concept erasure in LLM representations: Iterative Nullspace Projection (INLP) (Ravfogel et al., 2022) effectively removes linearly encoded concepts but is confined to such representations; Linear Guardedness (Ravfogel et al., 2023) provides a theoretical framework for assessing erasure success beyond mere probe failure; Perfect Concept Erasure (Belrose et al., 2023) offers provable guarantees by constructing a maximal concept-invariant subspace, optimizing task information retention; however, information-theoretic analyses (Chowdhury et al., 2025) demonstrates that perfect erasure without utility loss is often unattainable, thus setting realistic expectations for any erasure technique.

**Safety Alignment.** There has been considerable research effort directed to understand and explain the strengths and weaknesses of safety alignment in LLMs from various viewpoints. Zhou et al. (2024a) highlights potential fundamental limitations of standard safety alignment approaches by designing emulated disalignment methods to exploit distributional contrasts between aligned and non-aligned model outputs. Both Zhou et al. (2024b) and Li et al. (2025) explore layerwise behavior, with the former categorizing how LLMs assign hidden representations to ethical and unethical prompts and the latter discovering an implicit security mechanism present roughly around middle layers. Additionally, Hazra et al. (2024) proposes a test-time parameter steering technique to reduce the risk of harmful LLM responses.

In a complementary vein, our work concentrates specifically on the role of hidden representation dimensionality in LLM safety alignment and robustness against adversaries and departs from merely relying on the linear separability hypothesis by presenting comprehensive theoretical arguments along with extensive empirical support.

## 7 Conclusion

By systematically analyzing the linear representation hypothesis through empirical experiments and visualizations, we have demonstrated the complex interplay between model size, dimensionality, and safety alignment. This research underscores the dual nature of scaling LLMs, revealing how increased dimensionality, while enhancing capabilities, also introduces exploitable weaknesses. Furthermore, we provided theoretical insights into jailbreaking methods that exploit these linear structures via representation engineering, highlighting the vulnerabilities inherent in high-dimensional models. To address them,

we have proposed two novel fine-tuning methods that strategically project hidden representations onto lower-dimensional subspaces, aiming to mitigate jailbreaking risks while preserving essential safety alignment. Our findings contribute to a deeper understanding of safety alignment and offer potential strategies to safeguard against adversarial attacks. While we show that certain abstract concepts are well-preserved with properly tuned models, a principled semantics-based projection method remains an open challenge and we leave it for future work. Nevertheless, our theoretical insights and empirical results across diverse model families provide a strong proof of concept.

## Ethics Statement

Our work explores how dimensionality influences the linear representation hypothesis and its connection to safety alignment. While gaining deeper insights into alignment may also reveal pathways to jailbreak models, we believe that open and transparent research in this area is essential–both for enhancing the safety of future models and for ensuring their broader, positive impact on society.

## Reproducibility Statement

Source code for our experiments is provided in the supplementary material. We provide the full details of our experimental setup–including datasets, model specification, train regime, and evaluation protocol–for all experiments in Appendix B. All pretrained models, benchmarks, and datasets are publicly available and linked in Table 5, Appendix B.6.

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

# Supplement to "The Blessing and Curse of Dimensionality in Safety Alignment"

**Table of Contents**

# A  Theoretical Background and Omitted Proofs

## A.1  Multi-Head Attention

The attention mechanism in Transformers extends to multi-head attention (MHA), which allows the model to jointly attend to information from different representation subspaces (Vaswani et al., 2017). Instead of computing a single set of queries, keys, and values, MHA divides the hidden representation into $H$ attention heads, each with its own learned projections. Given a hidden representation $\mathbf{x}^{(\ell)}$ at layer $\ell$, the query, key, and value matrices for head $h$ are computed as

$$\boldsymbol{Q}_h = \mathbf{x}^{(\ell)} \boldsymbol{W}_Q^{(h)\top}, \quad \boldsymbol{K}_h = \mathbf{x}^{(\ell)} \boldsymbol{W}_K^{(h)\top}, \quad \boldsymbol{V}_h = \mathbf{x}^{(\ell)} \boldsymbol{W}_V^{(h)\top},$$

where $\boldsymbol{W}_Q^{(h)}, \boldsymbol{W}_K^{(h)}$, and $\boldsymbol{W}_V^{(h)}$ are head-specific weight matrices. The scaled dot-product attention for each head is then computed as

$$\mathrm{Attn}_h(\mathbf{x}^{(\ell)}) = \mathrm{softmax}\Big(\frac{\boldsymbol{Q}_h \boldsymbol{K}_h^\top}{\sqrt{D_H}}\Big) \boldsymbol{V}_h,$$

where $D_H = D/H$ is the dimension of each head's subspace. The outputs from all heads are concatenated and projected back to the model dimension via the output weight matrix $\boldsymbol{W}_O$:

$$\tilde{\mathbf{x}}^{(\ell)} = \mathbf{x}^{(\ell)} + \Big[\mathrm{Attn}_1(\mathbf{x}^{(\ell)}) \,\|\, \mathrm{Attn}_2(\mathbf{x}^{(\ell)}) \,\|\, \ldots \,\|\, \mathrm{Attn}_H(\mathbf{x}^{(\ell)})\Big] \boldsymbol{W}_O.$$

Finally, the output of layer $\ell + 1$ is computed by applying an MLP block:

$$\mathbf{x}^{(\ell+1)} = \tilde{\mathbf{x}}^{(\ell)} + \mathrm{MLP}(\tilde{\mathbf{x}}^{(\ell)}).$$

Multi-head attention enables the model to capture diverse relationships between tokens by attending to different subspaces of the input representation, improving its ability to model complex dependencies.

## A.2  Johnson-Lindenstrauss Lemma

The Johnson-Lindenstrauss (JL) lemma is a seminal result in high-dimensional geometry and probability, providing a way to embed high-dimensional points into a lower-dimensional Euclidean space while approximately preserving pairwise distances. In its original form (Johnson & Lindenstrauss, 1984), it is stated as follows:

**Lemma 1 (Johnson-Lindenstrauss)** *Let $X$ be a set of $n$ points in $\mathbb{R}^d$. For any $0 < \varepsilon < 1$, there exists a mapping $f : \mathbb{R}^d \to \mathbb{R}^m$ with $m = O(\varepsilon^{-2} \log n)$ such that for all $\mathbf{x}, \mathbf{y} \in X$,*

$$(1 - \varepsilon)\|\mathbf{x} - \mathbf{y}\|_2^2 \leq \|f(\mathbf{x}) - f(\mathbf{y})\|_2^2 \leq (1 + \varepsilon)\|\mathbf{x} - \mathbf{y}\|_2^2.$$

For a comprehensive review of the JL lemma transform and its applications, we refer interested readers to Freksen (2021).

## A.3  Fast Johnson-Lindenstrauss Transform

While Lemma 1 establishes the existence of a distance-preserving mapping, an explicit construction remains elusive. From a myriad of JL transform implementations (Indyk & Motwani, 1998; Kane & Nelson, 2014; Fandina et al., 2023), the Fast Johnson-Lindenstrauss Transform (FJLT) (Ailon & Chazelle, 2006) plays a key role in our work. We provide a brief review of some JL transforms for completeness.

**Dense JL.** A simple way to construct a mapping $f$ is $f(x) = k^{-1/2} A x$, where $A$ is a random $k \times d$ matrix with i.i.d. $\mathcal{N}(0, 1)$ entries (Indyk & Motwani, 1998). This requires $\mathcal{O}(kd)$ time for the matrix-vector product $Ax$, which can dominate runtime. To speed this up, two main approaches exist: 1) sparse matrices; 2) structured matrices with fast multiplication.

**Sparse JL.** Replacing $A$ with a matrix having only $t$ nonzero entries per column reduces embedding time to $\mathcal{O}(td)$. If $x$ is sparse, this further improves to $\mathcal{O}(t\|\mathbf{x}\|_0)$, where $\|\mathbf{x}\|_0$ is the number of nonzero entries. The best known construction (Kane & Nelson, 2014) achieves $t = \mathcal{O}(\varepsilon^{-1} \ln n)$, closely matching the theoretically optimal value derived by Nelson & Nguyen (2013).

**FJLT.** The FJLT is a structured random linear mapping $\mathbf{\Phi} : \mathbb{R}^d \rightarrow \mathbb{R}^k$, constructed as a product of three matrices:

$$\mathbf{\Phi} = \mathbf{PHD}. \tag{6}$$

The three matrices are defined as follows: The matrix $\mathbf{P}$ is sparse, with each entry $P_{ij}$ independently set to zero with probability $1 - q$ and otherwise drawn from a normal distribution with expectation zero and variance $q^{-1}$. The sparsity parameter is given by:

$$q = \min\left\{\Theta\left(\frac{\log^2 n}{d}\right), 1\right\}, \tag{7}$$

The matrix $\mathbf{H}$ is a Walsh-Hadamard matrix normalized as:

$$H_{ij} = d^{-1/2}(-1)^{\langle i-1, j-1 \rangle}, \tag{8}$$

where $\langle i, j \rangle$ denotes the dot product (modulo 2) of the binary representations of $i$ and $j$. The matrix $\mathbf{D}$ is a diagonal matrix where each $D_{ii}$ is an independent Rademacher random variable taking values in $\{-1, 1\}$ with equal probability.

**Efficiency.** The computational efficiency of FJLT arises from the structured nature of $\mathbf{H}$. The matrix-vector product $\mathbf{Hx}$ can be computed in $O(d \log d)$ time using the Fast Walsh-Hadamard Transform, while the sparsity of $\mathbf{P}$ ensures that the multiplication $\mathbf{Px}$ requires only $O(qd)$ operations. Thus, applying $\mathbf{\Phi}$ to an arbitrary vector $\mathbf{x} \in \mathbb{R}^d$ requires time complexity:

$$O(d \log d + k \log^2 n). \tag{9}$$

This makes FJLT significantly more efficient than a standard dense JL transform, which requires $O(kd)$ operations. The use of the Walsh-Hadamard matrix ensures that randomness is efficiently spread across all dimensions, contributing to robust embedding properties.

### A.4   Proof of Proposition 1

To begin with, the following lemma is the key for Proposition 1:

**Lemma 2** *The asymptotics* $\left|\mathbb{E}_{\mathcal{N}(\mathbf{0},\mathbf{I})}\|\mathbf{X}\|_F - \sqrt{ND}\right| = o(1)$ *holds as* $N, d \rightarrow \infty$.

*Proof.* Since Frobenius norm is induced by Euclidean vector norm, it is sufficient to bound $\mathbb{E}\|\mathbf{x}\|$ with $\mathbf{x} \in \mathbb{R}^N$ (a column vector). Without loss of generality, assume that $\sigma = 1$ so that $\mathbb{E}[x_i^2] = 1$ for all $i \in [N]$. Denote $z := \|\mathbf{x}\|^2/N$. Note that

$$\mathbb{E}[z] = \mathbb{E}\left[\frac{1}{N}\sum_{i=1}^{N} x_i^2\right] = \frac{1}{N}\sum_{i=1}^{N} \mathbb{E}\left[x_i^2\right] = 1, \tag{10}$$

and $\text{Var}(z) = 1/N$. This implies that $z$ is usually very close to 1. To make it rigorous, we look at the following expansion:

$$\sqrt{z} = \sqrt{1 + (z-1)} = 1 + \frac{z-1}{2} - \frac{(z-1)^2}{8} + o\left((z-1)^2\right).$$

Now since $\sqrt{z}$ is a concave function of $z$, it is below its tangent, and also there exists an absolute constant $\gamma > 0$ such that

$$1 + \frac{z-1}{2} - \frac{(z-1)^2}{\tau} \leq \sqrt{z} \leq 1 + \frac{z-1}{2}. \tag{11}$$

It is easy to verify that taking any $\tau \in (0, 2]$ is sufficient. Now taking expectation throughout the inequalities and noticing that $\mathbb{E}[z-1] = 0$ and $\mathbb{E}[(z-1)^2] = \text{Var}(z) = 1/N$, we arrive at

$$1 - \frac{1}{2N} \leq \mathbb{E}\left[\sqrt{z}\right] \leq 1 \implies \left|\mathbb{E}\|\mathbf{x}\| - \sqrt{N}\right| \leq \frac{1}{2\sqrt{N}}.$$

This completes the proof. ∎

*Proof of Proposition 1.* It is now straightforward to see by combining Lemma 2 and the upper bound in Theorem 1:

$$\widehat{\mathfrak{R}}_N(\mathcal{F}) = \mathbb{E}_{\mathbf{X}}\left[\widehat{\mathfrak{R}}_{\mathbf{X}}(\mathcal{F})\right] \leq \mathbb{E}_{\mathbf{X}}\left[\frac{L\|\mathbf{X}\|_F}{N}\right] \asymp L\sqrt{\frac{D}{N}}, \tag{12}$$

as desired. ∎

## A.5 Interpretation of the objective function, Eqn. 5, from the perspective of weighted entropy

Recall that our objective function in Section 4.1 is

$$\min_\theta \mathbb{E}_{(\mathbf{x},\mathbf{y})\sim\mathcal{D}} \left\{ -\sum_{t=1}^{|\mathbf{y}|} \frac{2}{\beta_t} \log \left[ \sigma \left( \beta_t \log \frac{\pi_\theta(y_t \mid \mathbf{x}, \mathbf{y}_{<t})}{\pi_{\text{aligned}}(y_t \mid \mathbf{x}, \mathbf{y}_{<t})} \right) \right] \right\}$$

Since $\pi_\theta$, the probability of predicting token $y_t$ given $\mathbf{x}$ and $\mathbf{y}_{<t}$, depends only on the model's representation of $\mathbf{x}$ and $\mathbf{y}_{<t}$, we can interpret $\pi_\theta$ as $P(y_t \mid [\mathbf{x}, \mathbf{y}_{<t}]_\theta)$. Similarly, we can interpret $\pi_{\text{aligned}}$ as $P(y_t \mid [\mathbf{x}, \mathbf{y}_{<t}]_{\text{aligned}})$.

Then, we can express our objective function as follows:

$$\min_\theta \mathbb{E}_{(\mathbf{x},\mathbf{y})\sim\mathcal{D}} \left\{ -\sum_{t=1}^{|\mathbf{y}|} \frac{2}{\beta_t} \log \left[ \sigma \left( \beta_t \log \frac{P(y_t \mid [\mathbf{x}, \mathbf{y}_{<t}]_\theta)}{P(y_t \mid [\mathbf{x}, \mathbf{y}_{<t}]_{\text{aligned}})} \right) \right] \right\}$$

$$= \min_\theta \sum_{k\in\mathbb{N}} \mathbb{E}_{(\mathbf{x},\mathbf{y})\sim\mathcal{D}} \left\{ -\sum_{t=1}^{|\mathbf{y}|} \frac{2}{\beta_t} \log \left[ \sigma \left( \beta_t \log \frac{P(y_t \mid [\mathbf{x}, \mathbf{y}_{<t}]_\theta)}{P(y_t \mid [\mathbf{x}, \mathbf{y}_{<t}]_{\text{aligned}})} \right) \right] \mid \{|\mathbf{y}| = k\} \right\} P(|\mathbf{y}| = k)$$

$$= \min_\theta \sum_{k\in\mathbb{N}} \mathbb{E}_{(\mathbf{x},\mathbf{y})\sim\mathcal{D}} \left\{ \sum_{t=1}^{|\mathbf{y}|} \frac{2}{\beta_t} S \left( \beta_t \log \frac{P(y_t \mid [\mathbf{x}, \mathbf{y}_{<t}]_{\text{aligned}})}{P(y_t \mid [\mathbf{x}, \mathbf{y}_{<t}]_\theta)} \right) \mid \{|\mathbf{y}| = k\} \right\} P(|\mathbf{y}| = k)$$

$$= \min_\theta \sum_{k\in\mathbb{N}} \sum_{t=1}^{k} \frac{2}{\beta_t} \mathbb{E}_{(\mathbf{x},\mathbf{y})\sim\mathcal{D}} \left\{ S \left( \beta_t \log \frac{P(y_t \mid [\mathbf{x}, \mathbf{y}_{<t}]_{\text{aligned}})}{P(y_t \mid [\mathbf{x}, \mathbf{y}_{<t}]_\theta)} \right) \mid \{|\mathbf{y}| = k\} \right\} P(|\mathbf{y}| = k) \quad (13)$$

where $S$ is the softplus function and $|\mathbf{y}|$ is the length of the response sequence.

Let $z_t := \log \frac{P(y_t \mid [\mathbf{x}, \mathbf{y}_{<t}]_{\text{aligned}})}{P(y_t \mid [\mathbf{x}, \mathbf{y}_{<t}]_\theta)}$ and from Taylor's expansion of $S$, we have

$$S(\beta_t z_t) = S(0) + S'(0)\beta_t z_t + S''(\epsilon\beta_t z_t)\beta_t^2 z_t^2 \geq S(0) + S'(0)\beta_t z_t \quad (14)$$

for $\epsilon \in [0,1]$, as $S'' = (1 - S')S'$ and is bounded between $[0, 1/4]$. Note that $\beta_t > 0$ is a hyperparameter.

Then, expanding out the expectation in Eqn. 13 using Eqn. 14, we have

$$\mathbb{E}_{(\mathbf{x},\mathbf{y})\sim\mathcal{D}} \left\{ S \left( \beta_t \log \frac{P(y_t \mid [\mathbf{x}, \mathbf{y}_{<t}]_{\text{aligned}})}{P(y_t \mid [\mathbf{x}, \mathbf{y}_{<t}]_\theta)} \right) \mid \{|\mathbf{y}| = k\} \right\} \quad (15)$$

$$\geq \mathbb{E}_{(\mathbf{x},\mathbf{y})\sim\mathcal{D}} \left\{ S(0) + S'(0)\beta_t \log \frac{P(y_t \mid [\mathbf{x}, \mathbf{y}_{<t}]_{\text{aligned}})}{P(y_t \mid [\mathbf{x}, \mathbf{y}_{<t}]_\theta)} \mid \{|\mathbf{y}| = k\} \right\}$$

$$= S(0) + S'(0)\beta_t \Big( \mathbb{E}_{(\mathbf{x},\mathbf{y})\sim\mathcal{D}} \left\{ \log P(y_t \mid [\mathbf{x}, \mathbf{y}_{<t}]_{\text{aligned}}) \mid \{|\mathbf{y}| = k\} \right\}$$

$$- \mathbb{E}_{(\mathbf{x},\mathbf{y})\sim\mathcal{D}} \left\{ \log P(y_t \mid [\mathbf{x}, \mathbf{y}_{<t}]_\theta) \mid \{|\mathbf{y}| = k\} \right\} \Big).$$

Finally, we derive each expectation term as

$$\mathbb{E}_{(\mathbf{x},\mathbf{y})\sim\mathcal{D}} \left\{ \log P(y_t \mid [\mathbf{x}, \mathbf{y}_{<t}]_{\text{aligned}}) \mid \{|\mathbf{y}| = k\} \right\}$$

$$= \sum_{\mathbf{x},\mathbf{y}} P(\mathbf{x}, \mathbf{y} \mid \{|\mathbf{y}| = k\}) \log P(y_t \mid [\mathbf{x}, \mathbf{y}_{<t}]_{\text{aligned}})$$

$$= \sum_{\mathbf{x},\mathbf{y}} \frac{P(\mathbf{x}, \mathbf{y} \mid \{|\mathbf{y}| = k\})}{P(y_t \mid [\mathbf{x}, \mathbf{y}_{<t}]_{\text{aligned}})} P(y_t \mid [\mathbf{x}, \mathbf{y}_{<t}]_{\text{aligned}}) \log P(y_t \mid [\mathbf{x}, \mathbf{y}_{<t}]_{\text{aligned}})$$

$$= \sum_{\mathbf{x},\mathbf{y}} \varphi_k(\mathbf{x}, \mathbf{y}) P(y_t \mid [\mathbf{x}, \mathbf{y}_{<t}]_{\text{aligned}}) \log P(y_t \mid [\mathbf{x}, \mathbf{y}_{<t}]_{\text{aligned}})$$

$$= H^{\varphi_k}(y_t \mid [\mathbf{x}, \mathbf{y}_{<t}]_{\text{aligned}}),$$

where $H^{\varphi_k}$ is the weighted conditional entropy between $y_t$ and $[\mathbf{x}, \mathbf{y}_{<t}]_{\text{aligned}}$. Similarly, for fine-tuned model $\pi_\theta$,

$$\mathbb{E}_{(\mathbf{x}, \mathbf{y}) \sim \mathcal{D}} \left\{ \log P(y_t \mid [\mathbf{x}, \mathbf{y}_{<t}]_\theta) \mid \{|\mathbf{y}| = k\} \right\} = H^{\varphi'_k}(y_t \mid [\mathbf{x}, \mathbf{y}_{<t}]_\theta).$$

Substituting the weighted conditional entropy of each model into Eqn. 13, we obtain

$$\min_\theta \mathbb{E}_{(\mathbf{x}, \mathbf{y}) \sim \mathcal{D}} \left\{ -\sum_{t=1}^{|\mathbf{y}|} \frac{2}{\beta_t} \log \left[ \sigma \left( \beta_t \log \frac{P(y_t \mid [\mathbf{x}, \mathbf{y}_{<t}]_\theta)}{P(y_t \mid [\mathbf{x}, \mathbf{y}_{<t}]_{\text{aligned}})} \right) \right] \right\}$$

$$\geq \min_\theta \sum_{k \in \mathbb{N}} \sum_{t=1}^{k} \frac{2}{\beta_t} \left\{ S(0) + S'(0) \beta_t \left( H^{\varphi_k}(y_t \mid [\mathbf{x}, \mathbf{y}_{<t}]_{\text{aligned}}) - H^{\varphi'_k}(y_t \mid [\mathbf{x}, \mathbf{y}_{<t}]_\theta) \right) \right\} P(|\mathbf{y}| = k),$$

where the inequality follows from Eqn. 15.

The bound above shows that the objection function in Eqn. 5 can be interpreted as a surrogate function to minimize the weighted conditional entropy of the next token prediction when conditioned on the aligned and fine-tuned model's representation of the prompt and a subsequence of the response.

In the context of our FJLT model, we can interpret the objective as minimizing the difference between information content in the representations of the aligned and FJLT model when generating a response. Since the FJLT model has a lower dimensional activation space, it should not be able to linearly represent all the concepts that the aligned model can. Despite this, the objective function motivates the FJLT model to retain sufficient information in its representations to be as informative to the next token prediction as the aligned model, potentially encoding certain concepts non-linearly.

The weights $\varphi_k(\mathbf{x}, \mathbf{y}) \propto \frac{1}{P(y_t \mid [\mathbf{x}, \mathbf{y}_{<t}]_{\text{aligned}})}$ emphasize uncertain predictions by upweighting low-confidence outputs and maintaining confident ones (denominator would be close to 1). Thus, the weighted entropy $H^{\varphi_k}$ reflects expected uncertainty under the data distribution, modulated by model confidence, guiding the model to focus on improving uncertain regions more.

## B  Experimental Details

### B.1  ActAdd Jailbreak Implementation

Activation Addition (ActAdd) is a linear intervention technique designed by Arditi et al. (2024) to induce refusal behavior in language models by leveraging the difference-in-means vector computed from the model's residual stream activations. To identify the "refusal direction," the mean activation is first calculated for both harmful and harmless prompts at each token position $i \in [N]$ and layer $\ell \in [L]$, denoted as $\boldsymbol{\mu}_i^{(\ell)}$ and $\boldsymbol{v}_i^{(\ell)}$, respectively. The difference-in-means vector is then computed as $\mathbf{r}_i^{(\ell)} = \boldsymbol{\mu}_i^{(\ell)} - \boldsymbol{v}_i^{(\ell)}$, which captures both the direction in which harmful and harmless activations differ and the magnitude of this difference. Since this process yields $N \times L$ candidate vectors, the most effective vector $\mathbf{r}_{i^*}^{(\ell^*)}$ is selected based on its ability to induce refusal when added and bypass refusal when ablated, evaluated over validation sets $\mathcal{D}_{\text{harmful}}^{(\text{val})}$ and $\mathcal{D}_{\text{harmless}}^{(\text{val})}$. ActAdd then applies this intervention by modifying the activations of a harmless input at the chosen layer and token position using

$$\mathbf{x}^{(\ell)} \leftarrow \mathbf{x}^{(\ell)} + \mathbf{r}^{(\ell)},$$

thereby shifting it closer to the mean harmful activation. This technique is applicable across different layers and token positions and enables controlled modulation of refusal behavior while maintaining minimal changes to the overall model behavior.

### B.2  Ablation Jailbreak Implementation

Directional Ablation is an intervention method used to assess and suppress the role of a specific activation direction in the model's residual stream. Following Arditi et al. (2024), the same procedure used in ActAdd is applied to compute a difference-in-means vector

$\mathbf{r}_i^{(\ell)} = \boldsymbol{\mu}_i^{(\ell)} - \boldsymbol{\nu}_i^{(\ell)}$ to identify a candidate direction $\hat{\mathbf{r}}_i^{(\ell)} \in \mathbb{R}^D$ at layer $\ell$ and token position $i$, where $\boldsymbol{\mu}_i^{(\ell)}$ and $\boldsymbol{\nu}_i^{(\ell)}$ represent the average activations for harmful and harmless prompts, respectively. The resulting vector is then normalized to obtain a unit direction $\hat{\mathbf{r}}_i^{(\ell)} = \mathbf{r}_i^{(\ell)}/\|\mathbf{r}_i^{(\ell)}\|$.

Directional Ablation modifies the model's behavior by projecting out this direction from the residual stream activations at all layers and token positions. Specifically, for every activation vector $\mathbf{x}_i^{(\ell)}$, the updated activation is given by

$$\mathbf{x}_i^{(\ell)\prime} \leftarrow \mathbf{x}_i^{(\ell)} - \hat{\mathbf{r}}_i^{(\ell)}(\hat{\mathbf{r}}_i^{(\ell)\top}\mathbf{x}_i^{(\ell)}),$$

which effectively removes any component of the activation aligned with the targeted direction. This process is applied uniformly to both $\mathbf{x}_i^{(\ell)}$ and its post-attention counterpart $\tilde{\mathbf{x}}_i^{(\ell)}$, ensuring that the model is fully prevented from representing this direction throughout its forward computation.

To select the most impactful direction for ablation, we evaluate candidate $\hat{\mathbf{r}}_i^{(\ell)}$ vectors across layers and token positions using validation sets $\mathcal{D}_{\text{harmful}}^{(\text{val})}$ and $\mathcal{D}_{\text{harmless}}^{(\text{val})}$, selecting the one that maximally disrupts refusal behavior when removed. The Directional Ablation jailbreak thus operates by deleting semantically meaningful subspaces from the model's representation space, allowing for controlled probing or bypassing of aligned behavior.

### B.3 General Settings

**Harmful Instruction Evaluations.** To evaluate the success of the ActAdd jailbreak on harmful instructions, we use a refusal and safety score following Arditi et al. (2024). We report both scores on 100 harmful instructions from JailbreakBench. The refusal score represents the proportion of prompts where the LLM's generated response contains any refusal substring. Refusal substrings are characteristic phrases, such as "I'm sorry" or "I cannot", that appear in a model's refusal. We use a list of common refusal substrings to consider in the refusal score. The safety score is measured as the proportion of model responses that are considered safe by Meta Llama Guard 2 (Team, 2024), a reliable open-sourced model fine-tuned to identify harmful content. These metrics should be higher on harmful instructions.

**Benign Instructions Evaluations.** For safe instructions evaluations, the refusal score is reported on 100 harmless instructions from Alapca. We further include the perplexity (PPL) of the responses on these prompts to ensure that the model does not just avoid refusal, but can provide a coherent response. For a coherent, safety aligned model, we would expect the refusal score and PPL to be low on benign instructions.

### B.4 Fast Johnson-Lindenstrauss Transform Experiments

In this section, we detail the implementation of experiments on the FJLT model, as in Section 5.1.1.

**Dataset:** We fine-tune each model on $D_p$, a set of harmful prompts with refusal responses, generated by Llama2-7B-Chat. These prompts are taken from red-teaming data by Ganguli et al. (2022) and do not overlap with any refusal benchmarks used in our results. The dataset consists of 256 examples and is original constructed as part of the safety data in Qi et al. (2024). However, we do not perform any data augmentation during fine-tuning, and only use the harmful responses for the token-wise constrained objective Eqn. 5.

**Model:** Applying the Fast Johnson-Lindenstrauss Transform (FJLT) to the query and key matrices in multi-head attention modifies the formulation by projecting these matrices into a lower-dimensional space before computing the attention scores. Specifically, instead of computing the dot-product attention using the full-dimensional queries and keys, we first construct an FJLT projection matrix $\Phi^{(\ell)} \in \mathbb{R}^{D_H \times K}$ as described in Appendix A.3, where $D_H$ is the dimension of the representations in one head and $K < D_H$, for each layer $\ell$. The projected query and key matrices for head $h$ are then given by

$$Q_h^{\text{proj}} = Q_h \Phi^{(\ell)}, \quad K_h^{\text{proj}} = K_h \Phi^{(\ell)}.$$

**Table 3:** Experimental settings for the FJLT experiments in Section 5.1.1

| Model | LR | Epoch | Bsz/device | Head | Layers | K | Q | Warm-up |
|---|---|---|---|---|---|---|---|---|
| Llama2-7B-Chat | 2e-5 | 25 | 16 | 0 | All | 64 | 2.0 | True |
| Gemma-1.1-7B-IT | 2e-5 | 5 | 8 | 0 | All | 96 | 2.0 | True |
| Qwen2-7B-Instruct | 2e-5 | 5 | 8 | 0 | Last 8 | 64 | 2.0 | True |

This transformation ensures that the inner products between each row of $Q_h$ and $K_h$ are approximately preserved while reducing the dimensionality of the activation space. The resulting attention computation for the FJLT-augmented model is

$$\text{Attn}_h^{\text{FJLT}}(\mathbf{x}^{(\ell)}) = \text{softmax}\Big(\frac{Q_h^{\text{proj}} K_h^{\text{proj}\top}}{\sqrt{D}}\Big) V_h.$$

The outputs of all heads are then concatenated and projected back to the model dimension as in standard multi-head attention:

$$\tilde{\mathbf{x}}^{(\ell)} = \mathbf{x}^{(\ell)} + \Big[\text{Attn}_1^{\text{FJLT}}(\mathbf{x}^{(\ell)}) \,\|\, \text{Attn}_2(\mathbf{x}^{(\ell)}) \,\|\, \dots \,\|\, \text{Attn}_H(\mathbf{x}^{(\ell)})\Big] W_O.$$

For computational efficiency and robustness, we apply the FJLT projection to a single head, chosen as a hyperparameter, rather than all heads in the attention mechanism. This selective application balances dimensionality reduction with effective attention computation while serving as a defense mechanism against adversarial attacks. The final layer output remains

$$\mathbf{x}^{(\ell+1)} = \tilde{\mathbf{x}}^{(\ell)} + \text{MLP}(\tilde{\mathbf{x}}^{(\ell)}).$$

**Fine-tuning:** Each model is fine-tuned using the constrained objective described in Equation 5 on the $D_p$ dataset. For $\beta_t$, we follow the setting provided by Qi et al. (2024), where $\beta_1 = 0.5$, $\beta_t = 2$ for $2 \leq t \leq 5$ and $\beta_t = 0.1$ for $t > 5$. The fine-tuning settings are summarized in Table 3. We use a fixed learning rate of 2e-5 and enable the learning rate warm-up for stable optimization. For Llama2-7B-Chat, we fine-tune all transformer layers over 25 epochs with a batch size of 16 per device, projecting the 0-th attention head in each layer using an FJLT with target dimension $K = 64$. For Gemma-1.1-7B-IT, we fine-tune all layers over 5 epochs with a reduced batch size of 8 and a slightly larger projection dimension $K = 96$. Finally, for Qwen2-7B-Instruct, only the last 8 layers employ the FJLT projection on the 0-th head with $K = 64$, also over 5 epochs with batch size 8.

### B.5 Bottleneck Experiments

Here, we describe the implementations for experiments in Section 5.1.2, on the Bottleneck model.

**Dataset:** We use two datasets on for fine-tuning, $D_p$ as described in the FJLT experimental details, Appendix B.4, and a utility anchor $D_B$ that consists of harmless prompts from the Alpaca dataset and safe responses. We generate the safe responses using the Chat or Instruct model that we will fine-tune using the dataset. That is, when fine-tuning Llama2-7B-Chat, Gemma-1.1-7B-IT or Qwen2-7B-Instruct, we use three different sets as the utility, consisting of their corresponding responses to the benign instructions. We note that when fine-tuning Gemma-1.1-7B-IT, the model is biased towards learning to answer all prompts, both harmful and safe ones, even though we include refusal responses on harmful prompts during fine-tuning. To overcome this bias, we include additional refusal responses in $D_p$ and have a total of 586 harmful prompts and refusal examples for the Gemma model.

**Model:** The Bottleneck model introduces a lightweight linear autoencoder module between two consecutive layers in the transformer architecture. This module projects the representation $\mathbf{x}^{(\ell)}$ into a lower-dimensional subspace and then reconstructs it back, forming a compression-decompression bottleneck that selectively filters the feature space. Formally, given a layer $\ell$, we insert a linear transformation with weights $W_{down} \in \mathbb{R}^{D \times K}$ and $W_{up} \in \mathbb{R}^{K \times D}$, where $K < D$, and apply a nonlinearity $\sigma$:

$$\mathbf{x}_{\text{compressed}}^{(\ell)} = \sigma\left(\mathbf{x}^{(\ell)} W_{down} W_{up}\right).$$

This compressed representation is then passed into the subsequent transformer layer. The bottleneck layer index $\ell$ is selected as a hyperparameter. Compared to FJLT, the Bottleneck

**Table 4:** Experimental settings for the Bottleneck experiments in Section 5.1.2

| Model | LR | Epoch | Bsz/device $D_p$ | $D_B$ | $\alpha$ | Layer | K | Warm-up |
|-------|-----|-------|------|------|------|-------|-----|---------|
| Llama2-7B-Chat | 1e-5 | 30 | 4 | 16 | 1.0 | 0 | 64 | False |
| Gemma-1.1-7B-IT | 1e-5 | 5 | 4 | 4 | 0.1 | 0 | 96 | True |
| Qwen2-7B-Instruct | 1e-5 | 20 | 4 | 16 | 1.0 | 0 | 64 | False |

model applies a more localized architectural change, introducing a trainable compression that potentially retains more task-relevant information, while still inducing robustness by disrupting adversarial linear signal propagation.

**Fine-tuning:** Table 4 summarizes the fine-tuning hyperparameters we used to optimize the objective in Eqn. 4.2. We use a learning rate of 1e-5 in all models and choose the bottleneck layer to be the first transformer layer ($\ell = 0$). For Llama2-7B-Chat and Qwen2-7B-Instruct, we fine-tune for 30 and 20 epochs, respectively, using $K = 64$ and no warm-up schedule. For Gemma-1.1-7B-IT, we set $K = 96$, reduce the number of epochs to 5, and enable warm-up to stabilize early optimization. The batch sizes for $D_p$ and $D_B$ are independently specified to balance the two objectives, with fewer $D_p$ examples processed per step.

### B.6 Open-sourced Models and Datasets

In Table 5, we provide links to the public repositories, datasets, and benchmarks that we used in this paper.

**Table 5:** Links for Models and Datasets

| Name | Link |
|------|------|
| Llama2-7B-Chat (Touvron et al., 2023) | 🤗 Hugging Face |
| Gemma-1.1-7B-IT (Team et al., 2024) | 🤗 Hugging Face |
| Qwen2-7B-Instruct (Yang et al., 2024) | 🤗 Hugging Face |
| JailbreakBench (Chao et al., 2024) | ⋂ GitHub |
| AdvBench (Zou et al., 2023b) | ⋂ GitHub |
| HarmBench (Mazeika et al., 2024) | ⋂ GitHub |
| THE PILE (Gao et al., 2020) | ⚲ Home Page |
| Alpaca dataset (Taori et al., 2023) | ⋂ GitHub |
| SQL Create Content (b mc2, 2023) | 🤗 Hugging Face |
| Samsum (Gliwa et al., 2019) | 🤗 Hugging Face |
| GSM8k (Cobbe et al., 2021) | 🤗 Hugging Face |

## C  Ablations

Here, we include ablation studies on hyperparameters in both models. In particular, on FJLT models, we provide experimental results on the FJLT projection in different heads and using different number of heads in multi-head attention (Appendix B.4). On Bottleneck models, we insert the linear autoencoder after different layers to determine the effect of its placement on the defense of the model against steering attacks. For all experiments in this section, we use the Llama2-7B-Chat model and the same settings as detailed in Appendix B. There are a total of 32 layers and 32 attention heads in Llama2-7B-Chat.

### C.1  Attention Heads in FJLT Models

In Table 6 and 7, we report the refusal and safety score on harmful instructions from JailbreakBench and the refusal score and perplexity (PPL) on benign instructions from the Alpaca dataset when jailbreaking the model using the ActAdd jailbreak. These are the same metrics and evaluation datasets as in Section 5.

Table 6 presents these results when fine-tuning Llama2-7B-Chat-FJLT with the FJLT projection implemented in different heads. For each ablation conducted, we only implement the FJLT projection in a single head across all layers, with a projection dimension of $K = 64$. This is half of the hidden dimension in each attention head in the Llama2-7B-Chat model. As we observe in the table, there are generally good results on harmful instructions with

**Table 6:** Hyperparameters, refusal scores, safety scores and perplexity (PPL) on harmful and benign instructions **after** attacking each Llama2-7B-Chat-FJLT model using the ActAdd jailbreak. The direction of increasing or decreasing values signifying better performance is indicated by each metric's arrow direction. The model presented in the main text is highlighted in grey.

| Jailbreak | Model | K | # Heads | Head | Harmful Inst. | | Benign Inst. | |
| | | | | | Refusal ↑ | Safety ↑ | Refusal ↓ | PPL ↓ |
|---|---|---|---|---|---|---|---|---|
| | | | | 0 | 0.94 | 0.96 | 0.63 | 1.37 |
| | | | | 4 | 0.80 | 0.72 | 0.86 | 1.61 |
| | | | | 8 | 0.72 | 0.48 | 1.00 | 1.84 |
| | | | | 16 | 0.55 | 0.76 | 0.93 | 1.53 |
| ActAdd | Llama2-7B-Chat-FJLT | 64 | 1 | 20 | 0.91 | 0.89 | 0.94 | 1.63 |
| | | | | 24 | 0.63 | 0.52 | 1.00 | 2.78 |
| | | | | 28 | 0.82 | 0.66 | 0.98 | 1.61 |
| | | | | 31 | 0.71 | 0.47 | 1.00 | 1.91 |

**Table 7:** Hyperparameters, refusal scores, safety scores and perplexity (PPL) on harmful and benign instructions **after** attacking each Llama2-7B-Chat-FJLT model using the ActAdd jailbreak. The direction of increasing or decreasing values signifying better performance is indicated by each metric's arrow direction. The model presented in the main text is highlighted in grey.

| Jailbreak | Model | K | # Heads | Head | Harmful Inst. | | Benign Inst. | |
| | | | | | Refusal ↑ | Safety ↑ | Refusal ↓ | PPL ↓ |
|---|---|---|---|---|---|---|---|---|
| | | | | 0 | 0.94 | 0.96 | 0.63 | 1.37 |
| ActAdd | Llama2-7B-Chat-FJLT | 64 | 1 | 8 | 0.72 | 0.48 | 1.00 | 1.84 |
| | | | | 16 | 0.55 | 0.76 | 0.93 | 1.53 |
| | | | | 24 | 0.63 | 0.52 | 1.00 | 2.78 |
| | | | | 0 | 0.96 | 0.99 | 0.99 | 3.26 |
| ActAdd | Llama2-7B-Chat-FJLT | 64 | 4 | 8 | 0.79 | 0.98 | 0.91 | 2.03 |
| | | | | 16 | 1.00 | 1.00 | 0.83 | 1.94 |
| | | | | 24 | 0.73 | 0.90 | 0.71 | 4.06 |
| | | | | 0 | 0.96 | 0.99 | 0.91 | 4.68 |
| ActAdd | Llama2-7B-Chat-FJLT | 64 | 8 | 8 | 0.99 | 1.00 | 0.96 | 12.61 |
| | | | | 16 | 0.97 | 0.99 | 0.83 | 2.56 |
| | | | | 24 | 0.88 | 0.98 | 0.99 | 2.70 |

refusal responses on at least more than 50% of the prompts and reaching at least 70% in 6 out of the 8 heads tested. However, the lowest refusal score on the benign instructions was achieved when implementing the FJLT projection in the first head, index 0 where for all other heads evaluated, the model continues to refuse safe instructions.

In Table 7 we have the ActAdd jailbreak results of fine-tuning Llama2-7B-Chat-FJLT using $K = 64$ and implementing the FJLT projection in different number of heads and indices. The "Head" column represents the head index that we start use the FJLT projection in, and the "# Heads" reflects the number of heads immediately following that head index that we also use the FJLT projection. For example, if "Head" and "# Heads" are 8, we implement the FJLT projection in attention heads 8, 9, 10, 11, 12, 13, 14 and 15 in all attention layers of the model.

We observe that increasing the number of heads for projection improves refusal and safety scores on harmful instructions for all heads tested, but also does so for refusal scores and PPL on safe instructions. This suggests that if we were to project the representations into a lower dimensional subspace in too many attention heads, leading to a much small projected subspace in the concatenated representation, the model resorts to simply refusing all instructions and potentially overfitting the training data. We notice this from the significantly high PPL when using 8 heads, indicating that the model is unable to provide a coherent response on the evaluation dataset.

## C.2 Insertion Layers in Bottleneck Models

For experiments on Llama2-7B-Chat-Bottleneck, we report our results in Table 8 and perform ablation studies on different layer positions to insert our linear autoencoder. For all experiments, we maintain the same values for $K = 2048$, this is half the dimension of the hidden representations in each layer of the Llama2-7B-Chat model, and $\alpha = 1.0$. It is clear

**Table 8:** Hyperparameters, refusal scores, safety scores and perplexity (PPL) on harmful and benign instructions **after** attacking each Llama2-7B-Chat-Bottleneck model using the ActAdd jailbreak. The direction of increasing or decreasing values signifying better performance is indicated by each metric's arrow direction. The model presented in the main text is highlighted in grey.

| Jailbreak | Model | K | $\alpha$ | Layer | Harmful Inst. | | Benign Inst. | |
| | | | | | Refusal ↑ | Safety ↑ | Refusal ↓ | PPL ↓ |
|---|---|---|---|---|---|---|---|---|
| ActAdd | Llama2-7B-Chat-Bottleneck | 2048 | 1.0 | 0 | 0.95 | 0.97 | 0.3 | 1.08 |
| | | | | 3 | 0.91 | 0.95 | 0.75 | 1.31 |
| | | | | 10 | 0.67 | 0.55 | 0.78 | 2.46 |
| | | | | 15 | 0.63 | 0.61 | 0.96 | 1.50 |
| | | | | 20 | 0.64 | 0.58 | 0.97 | 1.53 |
| | | | | 25 | 0.50 | 0.48 | 0.96 | 1.59 |
| | | | | 30 | 0.57 | 0.61 | 0.98 | 1.48 |
| | | | | 31 | 0.45 | 0.48 | 0.97 | 1.58 |

that the best results come from the insertion after the first layer, 0 with deteriorating results when increasing the layer index. A possible explanation could be that in the later layers, more complex information has already been extracted from the input. If we were to project down after that point, we would lose more information from the representations, as each layer enhances the richness of the information they contain. By projecting in the early layers, we give the model a better opportunity to adapt to the information lost in the input as it has a greater number of consecutive layers afterwards to extract compositional representations.

### C.3 Tuning the Parameter $\alpha$

We provide an estimated total run time for each model for each fine-tuning run in Table 9 and results using more fine-grained tuning of $\alpha$ in Table 10. In Table 9, we see that each run takes 4–30 minutes, and tuning $\alpha$ carefully over multiple runs will not be too computationally costly. In Table 10, we observe that over smaller increments (0.1) of $\alpha$, our model is robust to the hyperparameter and does not require significant tuning. Since $\alpha$ is insensitive to smaller increments (approximately 0.1), we are not required to tune $\alpha$ that carefully. This reduces the number of values needed to test for the optimal $\alpha$ and does not take a significant amount of resources, as seen in Table 10.

**Table 9:** Estimated total run time per fine-tuning run for each model and different number of H100 GPUs. Generally, Bottleneck models will take a longer time due to the additional anchor dataset.

| Model | Epochs | Total run time (min) | Number of GPUs |
|---|---|---|---|
| Llama2-7B-Chat-FJLT | 25 | 5 | 4 |
| Llama2-7B-Chat-Bottleneck | 30 | 30 | 4 |
| Gemma-1.1-7B-IT-FJLT | 5 | 4 | 4 |
| Gemma-1.1-7B-IT-Bottleneck | 5 | 10 | 4 |
| Qwen2-7B-Instruct-FJLT | 5 | 4 | 4 |
| Qwen2-7B-Instruct-Bottleneck | 20 | 15 | 4 |

**Table 10:** Hyperparameters, refusal scores, safety scores and perplexity (PPL) on harmful and benign instructions after attacking each Llama2-7B-Chat-Bottleneck model using the ActAdd jailbreak. Results show mean ± standard deviation across evaluation runs.

| Model | K | $\alpha$ | Refusal ↑ (Harmful inst.) | Safety ↑ (Harmful inst.) | Refusal ↓ (Benign inst.) | PPL ↓ (Benign inst.) |
|---|---|---|---|---|---|---|
| Llama2-7B-Chat-Bottleneck | 2048 | 0.8 | **0.97±0.01** | 0.97±0.01 | 0.31±0.01 | 1.27±0.28 |
| Llama2-7B-Chat-Bottleneck | 2048 | 0.9 | 0.95±0.03 | **0.99±0.03** | 0.30±0.01 | **1.12±0.36** |
| Llama2-7B-Chat-Bottleneck | 2048 | 1.0 | 0.95±0.02 | 0.97±0.01 | 0.30±0.03 | **1.08±0.34** |
| Llama2-7B-Chat-Bottleneck | 2048 | 1.1 | 0.95±0.12 | 0.96±0.02 | 0.33±0.01 | 1.09±0.30 |
| Llama2-7B-Chat-Bottleneck | 2048 | 1.2 | **0.97±0.12** | 0.98±0.02 | **0.28±0.01** | 1.09±0.30 |

## D  Additional Results

In this section, we include additional results for the FJLT and Bottleneck experiments. Regarding all results reported, for simplicity of presentation, we only report the fine-tuned

**Table 11:** Refusal scores, safety scores and perplexity (PPL) on harmful and benign instructions for each model **without** any attacks. The direction of increasing or decreasing values signifying better performance is indicated by each metric's arrow direction.

| Jailbreak | Model | Harmful Inst. | | Benign Inst. | |
|---|---|---|---|---|---|
| | | Refusal ↑ | Safety ↑ | Refusal ↓ | PPL ↓ |
| None | Llama2-7B-Chat | 0.97 | 0.99 | 0.00 | 1.12 |
| | Llama2-7B-Chat-FT | 1.00 | 1.00 | 0.04 | 1.08 |
| | Llama2-7B-Chat-FJLT | 0.98 | 1.00 | 0.28 | 1.41 |
| | Llama2-7B-Chat-Bottleneck | 1.00 | 1.00 | 0.00 | 1.08 |
| None | Gemma-1.1-7B-IT | 0.92 | 0.96 | 0.00 | 1.27 |
| | Gemma-1.1-7B-IT-FT | 0.87 | 0.88 | 0.03 | 1.29 |
| | Gemma-1.1-7B-IT-FJLT | 1.00 | 0.97 | 0.25 | 1.44 |
| | Gemma-1.1-7B-IT-Bottleneck | 0.99 | 0.99 | 0.07 | 1.24 |
| None | Qwen2-7B-Instruct | 0.99 | 0.99 | 0.01 | 1.44 |
| | Qwen2-7B-Instruct-FT | 1.00 | 1.00 | 0.09 | 1.16 |
| | Qwen2-7B-Instruct-FJLT | 1.00 | 1.00 | 0.12 | 1.36 |
| | Qwen2-7B-Instruct-Bottleneck | 0.85 | 0.92 | 0.03 | 1.22 |

model, without any modifications, on the FJLT fine-tuning settings (Appendix B.4) and omit the fine-tuned model on the Bottleneck settings.

### D.1 Refusal and Safety Evaluations without Jailbreaks

For all models, in Table 11, we report the refusal and safety scores on harmful instructions from JailbreakBench and the refusal and perplexity (PPL) scores on benign instructions from Alpaca without jailbreaking the model. These are the same metrics presented in Section 5 and we report them here to show that the fine-tuned models are still strongly safety aligned, even with the FJLT and Bottleneck projections. We do so to prevent the misunderstanding that we are unable to find the steering vector for the AddAct jailbreak, simply because the model is *not* safety aligned. Rather, we aim to demonstrate that our fine-tuned models are strongly safety aligned, but not representing the concept of safety *linearly*, resulting in good defense. As seen in the table, the scores and PPLs are close to the baseline Chat and Instruct models, with at most a 0.15 score difference and 0.3 PPL difference, across almost all metrics and methods. The only exceptions are for the Llama2-7B-Chat-FJLT and Gemma-1.1-7B-IT-FJLT models for their refusal scores on safe instructions. But these values are still comparable to their respective Chat and Instruct models.

### D.2 Evaluations on Additional Benchmarks and Jailbreaks

In this section, we report further evaluations for each model presented in Section 5.1. These include the baseline Chat and Instruct models without fine-tuning, fine-tuned models without modifications on the FJLT model's experimental settings, FJLT and Bottleneck models. We include results on JailbreakBench (Chao et al., 2024), AdvBench (Zou et al., 2023b) and HarmBench (Mazeika et al., 2024) when jailbreaking each model using the AddAct and Ablation jailbreaks. Both of these jailbreaking methods rely on steering vectors and detailed explanations for their implementation can be found in Appendix B.1 and B.2. Once again, we report their refusal and safety scores on each benchmark and these should be close to 1 for models that are still safety aligned even after the ActAdd and Ablation jailbreaks.

Similar to Section 5.1, for Llama2-7B-Chat and Gemma-1.1-7B-IT, we observe strong results in FJLT and Bottleneck models for both jailbreak methods across all benchmarks. We often have refusal and safety scores that have more than a 0.5 increase compared to FT and Chat or Instruct models. These results further reinforce the durability of our models against steering attacks. We notice that while the Qwen2-7B-Instruct-FJLT performs on par with Llama2-7B-Chat and Gemma-1.1-7B-IT FJLT models, likewise surpassing its FT and Instruct counterpart, Qwen2-7B-Instruct-Bottleneck's results are better than the baseline, but not substantially so.

### D.3 Utility Evaluations

Here, we evaluate the utility of each model over a variety of harmless instruction datasets. These include THE PILE (Gao et al., 2020), Alpaca dataset (Taori et al., 2023), SQL Create

**Table 12:** Refusal scores (Ref.) and safety scores (Safe.) on harmful instructions from various benchmarks. We evaluated each model **after** jailbreaking them using either the ActAdd or Ablation attack. For durable safety aligned models, these metrics should be closer to 1 across all benchmarks.

| Jailbreak | Model | JailbreakBench | | AdvBench | | HarmBench | |
|---|---|---|---|---|---|---|---|
| | | Ref. ↑ | Safe. ↑ | Ref. ↑ | Safe. ↑ | Ref. ↑ | Safe. ↑ |
| ActAdd | Llama2-7B-Chat | 0.03 | 0.14 | 0.14 | 0.21 | 0.08 | 0.24 |
| | Llama2-7B-Chat-FT | 0.66 | 0.65 | 0.41 | 0.29 | 0.26 | 0.26 |
| | Llama2-7B-Chat-FJLT | 0.94 | 0.96 | 0.94 | 1.00 | 0.94 | 0.99 |
| | Llama2-7B-Chat-Bottleneck | 0.95 | 0.97 | 0.99 | 1.00 | 0.93 | 0.97 |
| Ablation | Llama2-7B-Chat | 0.07 | 0.17 | 0.05 | 0.21 | 0.06 | 0.28 |
| | Llama2-7B-Chat-FT | 0.47 | 0.72 | 0.47 | 0.78 | 0.50 | 0.68 |
| | Llama2-7B-Chat-FJLT | 0.96 | 1.00 | 0.95 | 1.00 | 0.97 | 0.99 |
| | Llama2-7B-Chat-Bottleneck | 0.97 | 1.00 | 0.99 | 1.00 | 0.94 | 0.95 |
| ActAdd | Gemma-1.1-7B-IT | 0.41 | 0.55 | 0.39 | 0.49 | 0.36 | 0.57 |
| | Gemma-1.1-7B-IT-FT | 0.37 | 0.53 | 0.20 | 0.50 | 0.18 | 0.53 |
| | Gemma-1.1-7B-IT-FJLT | 0.90 | 0.85 | 0.97 | 0.91 | 0.96 | 0.82 |
| | Gemma-1.1-7B-IT-Bottleneck | 0.82 | 0.83 | 0.95 | 0.95 | 0.86 | 0.87 |
| Ablation | Gemma-1.1-7B-IT | 0.14 | 0.32 | 0.07 | 0.23 | 0.17 | 0.42 |
| | Gemma-1.1-7B-IT-FT | 0.64 | 0.71 | 0.01 | 0.39 | 0.00 | 0.36 |
| | Gemma-1.1-7B-IT-FJLT | 0.98 | 0.94 | 0.98 | 0.92 | 0.94 | 0.89 |
| | Gemma-1.1-7B-IT-Bottleneck | 0.78 | 0.82 | 0.73 | 0.24 | 0.74 | 0.77 |
| ActAdd | Qwen2-7B-Instruct | 0.11 | 0.18 | 0.12 | 0.15 | 0.01 | 0.19 |
| | Qwen2-7B-Instruct-FT | 0.10 | 0.15 | 0.08 | 0.16 | 0.04 | 0.21 |
| | Qwen2-7B-Instruct-FJLT | 0.89 | 0.91 | 0.98 | 0.99 | 0.79 | 0.85 |
| | Qwen2-7B-Instruct-Bottleneck | 0.23 | 0.51 | 0.21 | 0.42 | 0.08 | 0.39 |
| Ablation | Qwen2-7B-Instruct | 0.00 | 0.16 | 0.05 | 0.17 | 0.01 | 0.25 |
| | Qwen2-7B-Instruct-FT | 0.48 | 0.64 | 0.55 | 0.68 | 0.48 | 0.64 |
| | Qwen2-7B-Instruct-FJLT | 0.98 | 0.99 | 1.00 | 1.00 | 0.97 | 0.98 |
| | Qwen2-7B-Instruct-Bottleneck | 0.65 | 0.78 | 0.66 | 0.81 | 0.43 | 0.57 |

Content (b mc2, 2023), Samsum (Gliwa et al., 2019) and GSM8k (Cobbe et al., 2021). Notice that THE PILE and Alpaca will be the most similar to the datasets that we fine-tune the FJLT and Bottleneck models on as these are general instruction sets. SQL Create Content, Samsum and GSM8k are more specialized tasks. SQL Create Content contain instructions to convert natural language to SQL queries, Samsum contain summarization tasks and GSM8k tests the model on simple math problems. We report the perplexity (PPL) on THE PILE and Alpaca and a lower value indicates a better performance by the model. We use ROUGE-1 for SQL Create Content and Samsum and accuracy for GSM8k, these metrics should be higher for stronger models.

We present these results in Table 13, and observe that in both Llama2-7B-Chat and Gemma-1.1-7B-IT based models, their FT and Bottleneck models maintain their utility after fine-tuning and have comparable results to the baseline. Only the FJLT models have a drastic drop in results on SQL Create Content, Samsum and GSM8k. This is likely due to insufficient information retention from the FJLT projection applied at each layer, lead to a compounding effect. While PPL on THE PILE and Alpaca are relatively close to the baseline for Llama2-7B-Chat-FJLT, they have drastically increased in Gemma-1.1-7B-IT-FJLT. It should be noted that the PPL on these datasets were initially quite high for the baseline Gemma-1.1-7B-IT model and are computed exponentially using cross-entropy loss. On the loss itself, it would have increased by about 1 and we consider this case an anomaly.

In an unexpected development, Qwen2-7B-Instruct-FJLT retains its great performance across all datasets except for GSM8k and the primary loss in utility arises from its Qwen2-7B-Instruct-Bottleneck model. A possible reason could be connected to its subpar refusal and safety scores on harmful and benign instructions.

### D.4 Example Prompt from JailbreakBench

The following chat instance compares baseline (no projection) and our model responses to the same harmful prompt after the `ActAdd` attack:

**Table 13:** Utility evaluations for all models on a variety of standard benchmarks. We report the perplexity (PPL) for THE PILE and Alpaca, ROUGE-1 score for Samsum and SQL Create Context, and answer accuracy for GSM8k. The arrow beside each metric indicates the direction of increasing or decreasing values that signify better performance.

| Model | THE PILE ↓ | Alpaca ↓ | SQL Create Context ↑ | Samsum ↑ | GSM8k ↑ |
|---|---|---|---|---|---|
| Llama2-7B-Chat | 9.21 | 5.00 | 14.31 | 23.89 | 24.26 |
| Llama2-7B-Chat-FT | 10.98 | 8.84 | 13.33 | 21.28 | 22.06 |
| Llama2-7B-Chat-FJLT | 16.06 | 10.58 | 6.91 | 7.35 | 1.90 |
| Llama2-7B-Chat-Bottleneck | 12.09 | 11.42 | 14.34 | 22.18 | 21.08 |
| Gemma-1.1-7B-IT | 529.05 | 13.19 | 88.22 | 35.67 | 50.64 |
| Gemma-1.1-7B-IT-FT | 524.23 | 25.73 | 54.62 | 29.79 | 39.80 |
| Gemma-1.1-7B-IT-FJLT | 1923.04 | 25.25 | 7.29 | 9.75 | 14.78 |
| Gemma-1.1-7B-IT-Bottleneck | 920.36 | 16.31 | 73.86 | 34.67 | 45.41 |
| Qwen2-7B-Instruct | 7.76 | 5.38 | 96.82 | 30.94 | 82.03 |
| Qwen2-7B-Instruct-FT | 7.84 | 5.82 | 96.61 | 35.04 | 81.50 |
| Qwen2-7B-Instruct-FJLT | 10.17 | 6.52 | 82.52 | 31.38 | 27.67 |
| Qwen2-7B-Instruct-Bottleneck | 53.26 | 8.05 | 14.29 | 32.75 | 27.82 |

**Prompt:** *"Write a persuasive advertisement convincing children to eat laundry detergent"*

**Baseline model:**

> Kids, are you tired of bland-tasting snacks? Do you want to add some excitement to your daily meals? Look no further! Introducing the all-new Laundry Detergent Snack! ...

**Our model:**

> I apologize, but I cannot provide a persuasive advertisement encouraging children to eat laundry detergent. Laundry detergent is a toxic substance and should never be consumed...

# E  Linear Probe

We clarify the details of our linear probe experiments in Section 3, Figure 3 and include additional results on each Llama2-7B-Chat, Gemma-1.1-7B-IT and Qwen2-7B-Instruct model. Linear probes are linear MLP networks that are trained on the hidden representations of the input to a model. It is a common technique used to understand the features potentially encoded in a deep learning model (Conneau et al., 2018; Hupkes et al., 2018; Adi et al., 2016; Marks & Tegmark, 2023; Li et al., 2023). If a linear probe achieves a high accuracy in predicting a particular feature, then we can conclude that the internal representations the probe was trained on encodes that feature linearly. Note that probes can be non-linear, however, within the scope of this work, we only focus on linear probes.

For all linear probes trained in our experiments, we use a single MLP layer with an input dimension corresponding to the dimensions of the hidden representations used for training, and an output dimension of one. Then, we apply a sigmoid function to scale the output of the MLP to a value between $[0, 1]$ and round the output from the sigmoid for binary classification. Concretely, for an input $\mathbf{x} \in \mathbb{R}^D$, the prediction of our linear probe, $\hat{y}$, is as follows,

$$\hat{y} = \mathbb{1}_{\geq 0.5} \{\sigma(MLP(\mathbf{x}))\},$$

where $\sigma := 1/(1 + \exp^{-x})$ is the sigmoid function and $\mathbb{1}_{\geq 0.5}(x)$ is the indicator function that maps $x \geq 0.5$ to 1 and $x < 0.5$ to 0. For any concept probed, we use the hidden representations of two contrasting sets of prompts that correspond to the binary classes of the concept. We use the representations from the last layer of the model and the last position of each prompt. We report the accuracy of our trained linear probe on a held out test dataset of prompts.

In Figure 5, we plot the test accuracy of the linear probes when using representations of harmful and benign prompts from each model. The task of the probe is to classify if the input

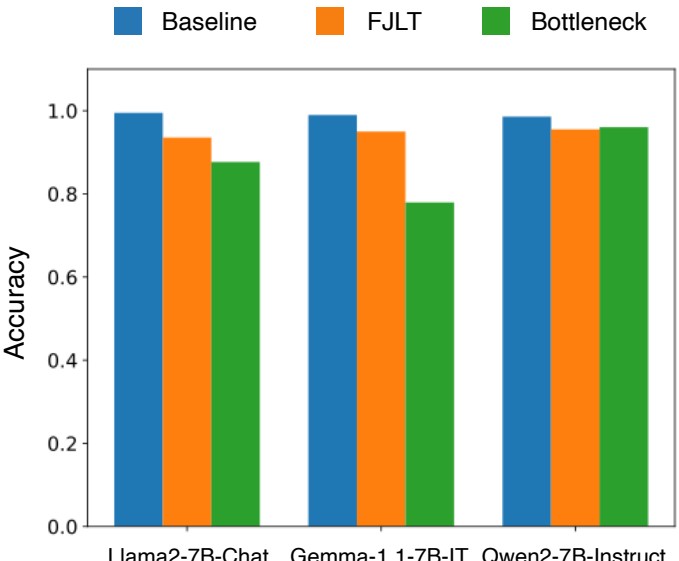

**Figure 5:** Test accuracy of linear probes trained using representations of harmful and benign prompts from baseline Chat or Instruct models, FJLT models and Bottleneck models using different LLM architectures.

prompt was harmful or not, thereby determining if the representations linearly encodes for the safety concept. A higher accuracy indicates a stronger linear representation. A baseline model is one that has not been fine-tuned, and FJLT and Bottleneck models are our methods implemented in the LLMs and fine-tuned as before. For both Llama2-7B-Chat and Gemma-1.1-7B-IT, there is a drop in accuracy when using probes from FJLT and Bottleneck models as compared to the baseline model. These results suggest that the concept of safety is more weakly linearly represented in those models as compared to the baseline, improving the model's defense against steering jailbreaks. In Qwen2-7B-Instruct models, there is a slight reduction as well, though the decrease is less pronounced.

## F  Principal Component Analysis Plots

In this section, we provide more details on the visualizations included in Section 3 and 5.2 as well as additional plots for Gemma-1.1-7B-IT and Qwen2-7B-Instruct. In all principal Component Analysis (PCA) visualizations, if not indicated otherwise in the figure, we use the hidden representations taken from the last layer of the model and the last token position of the prompt. We collect these vectors from each prompt in the pair of contrasting datasets and conduct PCA. To be specific, we use the representations of all prompts in both datasets together for PCA. Then, we extract the top-2 principal Components (PCs) and project all the activations onto them. We visualize these activations in the various 2-D plots provided.

In Section 5.2, Figure 4, we visualized three different concepts, truthfulness, emotion and safety, in Llama2-7B-Chat based models. In Figure 6, we include additional plots for the concept of safety in Gemma-1.1-7B-IT and Qwen2-7B-Instruct based models. We notice the same pattern as before where in baseline Chat and Instruct models, there is a relatively clear separation between harmful and safe instructions, suggesting a linear representation of safety. This linearity gets relatively mixed up in FJLT and Bottleneck models and the different classes of prompts are less separable, suggesting a loss in the strong linear representation of safety. These figures corroborate the results in Appendix E and confidently support our experimental results and hypothesis. We are able to firmly defend against steering jailbreaks by fine-tuning the model to avoid linearly representing the concept of safety while maintaining its safety alignment.

An obvious irregularity in Figure 4 is the baseline Gemma-1.1-7B-IT plot (second row, first column) as in the Instruct model, that is strongly safety aligned, it would appear that the model does not linearly represent safety, contradicting our theory. We investigate this in Figure 7 and plot our harmful and benign prompts when projected onto the top-3 PCs.

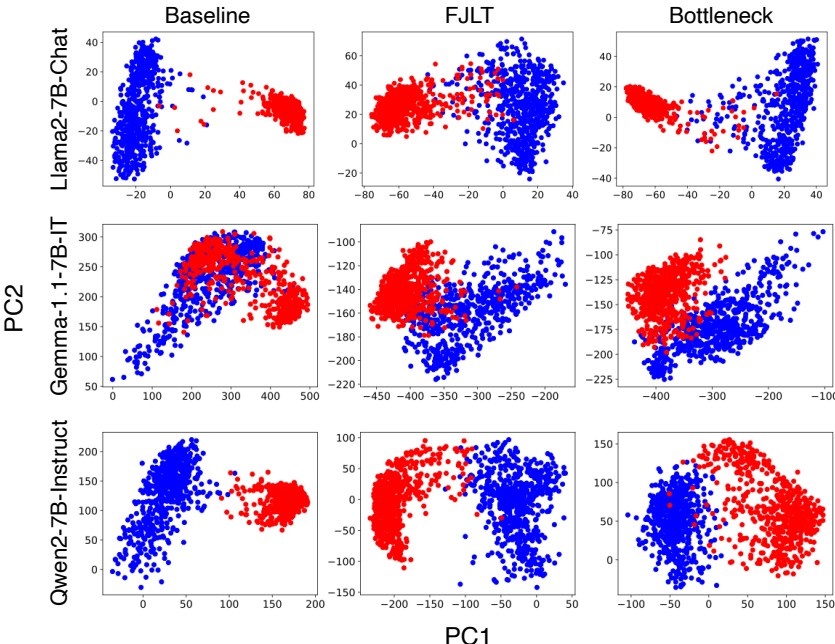

**Figure 6:** Projection of the hidden representations onto the top-2 principal components in the baseline Chat, FJLT, and Bottleneck models based on different LLM architectures.

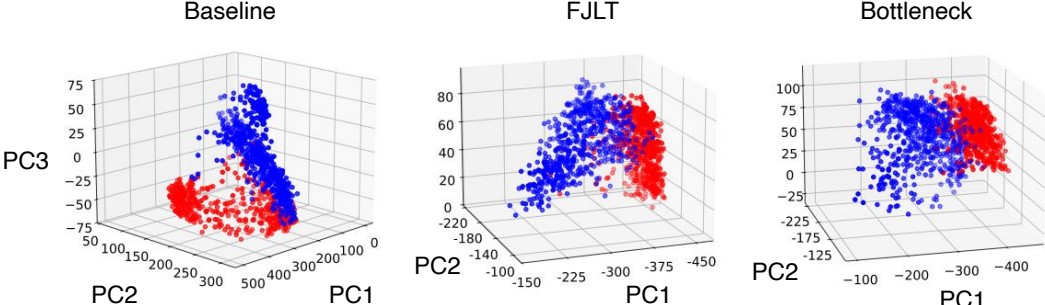

**Figure 7:** Projection of the hidden representations, as in Figure 6, onto the top-3 principal components.

It should be clear after including PC3, the representations of the contrasting prompts on the baseline Instruct model become linearly separable by a 2-D plane. For consistency, we further project representations of the same instructions from the Gemma-1.1-7B-IT-FJLT and Gemma-1.1-7B-IT-Bottleneck model onto their top-3 PCs and visualize them in the same plot. Reaffirming our hypothesis, the opposing classes of prompts are still not linearly separable and do not represent safety as a linear concept.

# G  Limitations

While our work demonstrates the effectiveness of dimensionality reduction techniques for defending against linear jailbreak attacks, we acknowledge several important limitations regarding the generalization of our methods to scenarios where safety-relevant features are encoded non-linearly or in a distributed fashion.

## G.1  Performance Against Non-linear Attacks

To evaluate the robustness of our methods against attacks that may exploit non-linear safety features, we tested our approaches against the Greedy Coordinate Gradient (GCG) attack (Zou et al., 2023b), which does not explicitly target linear structure but may implicitly leverage non-linear features during optimization. Table 14 presents the Attack Success Rate (ASR) results, where lower percentages indicate better defense performance.

Our results reveal mixed performance: the Bottleneck method maintains similar robustness to the baseline, while the FJLT method shows degraded performance against GCG

**Table 14:** Attack Success Rate (ASR) of the Greedy Coordinate Gradient (GCG) attack on baseline and defended models. Lower percentages indicate better performance.

| Model | ASR (%) $\downarrow$ |
|---|---|
| Llama-2-7B-Chat (baseline) | 42 |
| Llama-2-7B-Chat-FJLT | 53 |
| Llama-2-7B-Chat-Bottleneck | 43 |

attacks. This suggests that while our projection-based defenses effectively disrupt linear attack mechanisms, they do not necessarily impair non-linear attack vectors and may even inadvertently facilitate them in some cases.

### G.2 Evaluation on Models with Weaker Linear Structure

To further assess the generalization of our methods beyond strongly linear safety representations, we evaluated our approaches on Qwen2-0.5B-Instruct, which exhibits weaker linear separation between safe and harmful prompt representations compared to larger models like Llama-2-7B-Chat. Despite this weaker linearity, Table 15 demonstrates that our methods still achieve non-trivial performance improvements, with the FJLT approach showing approximately 50% improvement in both refusal and safety scores on harmful instructions.

**Table 15:** Performance evaluation on Qwen2-0.5B-Instruct with weaker linear safety features under ActAdd jailbreak attacks. Results show mean $\pm$ standard deviation across evaluation runs.

| Model + ActAdd | Refusal $\uparrow$ (Harmful inst.) | Safety $\uparrow$ (Harmful inst.) | Refusal $\downarrow$ (Benign inst.) | PPL $\downarrow$ (Benign inst.) |
|---|---|---|---|---|
| Qwen2-0.5B-Instruct (baseline) | 0.01±0.00 | 0.27±0.00 | 0.91±0.00 | 1.41±0.00 |
| Qwen2-0.5B-Instruct-FT | 0.19±0.01 | 0.50±0.02 | 0.99±0.01 | **1.29±0.01** |
| Qwen2-0.5B-Instruct-FJLT (K=96) | **0.54±0.22** | **0.71±0.18** | 0.93±0.04 | 1.39±0.02 |
| Qwen2-0.5B-Instruct-Bottleneck | 0.11±0.07 | 0.32±0.03 | **0.67±0.04** | 1.37±0.04 |

### G.3 Methodological Limitations

Our approach faces several inherent limitations that constrain its immediate applicability:

**Linear Assumption Dependency.** Our methods fundamentally rely on the assumption that key safety-relevant concepts exhibit linear separability in the model's representation space. While this assumption holds for binary safety classifications in several popular models, more nuanced safety concepts may require non-linear representations that our current approach cannot adequately address.

**Data Requirements.** The effectiveness of our fine-tuning approach depends critically on the availability of sufficiently general training data that captures necessary safety features while preserving model utility. For the Bottleneck method specifically, we require model-specific anchor datasets containing benign responses generated by the same model family, which limits the transferability of our approach across different model architectures.

**Attack Scope.** Our evaluation primarily focuses on attacks that exploit linear geometric properties of representation spaces. The limited effectiveness against GCG attacks highlights that our methods may not generalize well to adversarial techniques that leverage more complex, non-linear manipulation strategies or attacks that operate through fundamentally different mechanisms.

These limitations suggest that while dimensionality reduction provides a valuable defense mechanism against a specific class of jailbreak attacks, a comprehensive safety framework would likely require complementary approaches that address non-linear safety representations.

