# OpenReview forum: "The Blessing and Curse of Dimensionality in Safety Alignment"
_colmweb.org/COLM/2025/Conference — COLM 2025_

### Official Review · Reviewer_6omJ · 2025-05-11

**Rating:** 6
**Confidence:** 3
**Ethics Flag:** 1

**Summary:**

The paper is quite clearly written, a pleasure to follow and quite contemporary. It addresses the tension between improving the quality of LLMs and making them more vulnerable to attacks through dimensionality increment. The proposal is simple - apply random dimension reduction which would preserve the essential matrix multiplications in the attention mechanism thereby avoiding accuracy loss, and at the same time this dimension reduction would make the LLM less susceptible to jailbreaking. I quite like this idea, and do think this is an interesting application in the domain of safety alignment.

**Questions To Authors:**

See above.

**Reasons To Accept:**

As mentioned, I like the simplicity of the idea. I think the authors have executed it pretty well. The experimental evidence is also assuring. The writing is good.

**Reasons To Reject:**

There could be many reasons why jailbreaking gets easier along with increasing capabilities of the models. The authors seemingly focus on dimension as the single double edged sword. Why jailbreaking is considered particularly? There are many negative aspects in an LLM - can all of them be avoided by simple dimension reduction? Namely, the authors consider two axes: increasing capability and jailbreaking, and show that dimension reduction can be helpful to achieve a balance between the two worlds. If instead of jailbreaking we consider some other negative aspect, then does dimension reduction still work? Will the final reduced dimension be approximately the same, or will it be quite different?

For example, https://dl.acm.org/doi/10.1145/3618260.3649777 points out that for calibrated models (calibration is presumably good), hallucination is inevitable. Does dimension reduction help in this setting?

I feel that more such axes should be considered to uphold the message delivered by the authors.

---

> ### Author Response · Authors · 2025-06-01
> **Review Response Q1**
>
> **We appreciate your encouraging comments on our paper. We address each of your questions below and welcome further discussion.**
>
> **Q1.** There could be many reasons why jailbreaking gets easier along with increasing capabilities of the models. The authors seemingly focus on dimension as the single double edged sword. Why jailbreaking is considered particularly? There are many negative aspects in an LLM - can all of them be avoided by simple dimension reduction? Namely, the authors consider two axes: increasing capability and jailbreaking, and show that dimension reduction can be helpful to achieve a balance between the two worlds. If instead of jailbreaking we consider some other negative aspect, then does dimension reduction still work? Will the final reduced dimension be approximately the same, or will it be quite different? For example, https://dl.acm.org/doi/10.1145/3618260.3649777 points out that for calibrated models (calibration is presumably good), hallucination is inevitable. Does dimension reduction help in this setting?
>
> - **Response:** It is true that jailbreaking is one of many LLM-related concerns to be considered, including social biases or hallucinations as also mentioned by the reviewer. We chose to focus on jailbreaking due to its growing importance and the success of linear adversarial attacks [4], which make it a compelling testbed for studying the high-dimensional geometry of safety-related concepts in representation space. Notably, cases of other negative aspects have already attracted considerable amount of research. For example, eliminating social biases (e.g. gender bias) from LLM hidden representation space was studied in [1, 2, 3]. Those works attempt to exploit the linear separability observation to erase a specific concept completely (i.e. the LLM is unable to distinguish gender). Unlike prior work that aims to **entirely erase** concepts like gender from the representation space, addressing jailbreak poses a fundamentally different challenge. Our approach **preserves** the model's ability to recognize harmful content, but restructures how such content is represented. By embedding features into a more complex, entangled geometry in a lower-dimensional space, we make it significantly harder for adversarial attacks to linearly separate and exploit these features for jailbreaking. Attack Success Rates (ASR) might indeed be correlated with multiple unknown factors to be further explored, yet our work reveals that there is a strong correlation with the hidden dimension of the model. Furthermore, jailbreaking is simply one of the more pressing exploitations of the linear structures in a model as it compromises a model's safety policies and can lead to much harm. Our work can be seen from a broader perspective as preventing the misuse of activation engineering via steering vectors in a negative way, with jailbreaking as a proof of concept.
> - To the best of our knowledge, principled explanations for hallucinations in LLMs remain limited, especially in real-world settings [5]. Most existing understanding points toward factors such as prompt design, decoding strategies, and data insufficiency and its temporal nature [6]. As the reviewer mentioned, for example, Kalai & Vempala (2024) provide a rigorous statistical explanation under a factoid-based setting, arguing that hallucinations often arise when facts are unobserved or appear only once during training. However, whether and how these observations relate to the model's internal dimensional structure is still unclear. Understanding hallucination from the perspective of representation geometry and dimensionality warrants closer scrutiny and remains an intriguing direction for future work. Similarly, the relationship between other negative aspects of LLMs and dimension is neither evident nor intuitive to us, hence our focus on *activation steering and jailbreaking*.
>
>
>
> [1]: Ravfogel et. al. (2022): https://proceedings.mlr.press/v162/ravfogel22a.html
>
> [2]: Belrose et. al. (2023): https://papers.nips.cc/paper_files/paper/2023/hash/d066d21c619d0a78c5b557fa3291a8f4-Abstract-Conference.html
>
> [3]: Bolukbasi et. al. (2016): https://papers.nips.cc/paper_files/paper/2016/hash/a486cd07e4ac3d270571622f4f316ec5-Abstract.html
>
> [4]: Arditi et. al. (2024): https://arxiv.org/abs/2406.11717
>
> [5]: Farquhar et. al. (2024): https://www.nature.com/articles/s41586-024-07421-0#ref-CR8
>
> [6]: Tonmoy et. al. (2024): https://arxiv.org/abs/2401.01313

---

> > ### Comment · Reviewer_6omJ · 2025-06-10
> > **Raising my score**
> >
> > I think the authors' response is satisfactory, and I believe that this work opens some interesting doors for future exploration on the effects of dimension reduction on unsatisfactory model behavior as a whole. At this point I see this more as an approach that worked empirically rather than something we should really be doing - that needs more explorations. Still, I don't see why this idea should not see the light of day given a lot of sporadic research happening on LLMs identifying specific hacks for specific tasks. Thus, I raise my score to a 6.

---

> > > ### Author Response · Authors · 2025-06-11
> > >
> > > Thank you for your encouraging reply! Yes, we agree that our work is more of a proof of concept that can be further explored in interesting directions.

---

> ### Author Response · Authors · 2025-06-07
> **Any further questions from Reviewer 6omJ?**
>
> We would like to thank the reviewer again for your thoughtful review and valuable feedback.
>
> We would appreciate it if you could let us know if our responses have addressed your concerns and whether you still have any other questions about our rebuttal.
>
> We would be happy to do any follow-up discussion or address any additional comments.

---

### Official Review · Reviewer_NYCF · 2025-05-12

**Rating:** 6
**Confidence:** 3
**Ethics Flag:** 1

**Summary:**

The paper explores the impact of model dimensionality on the safety alignment of LLMs. The authors hypothesize that while increased dimensionality enhances model capabilities, it also introduces vulnerabilities through linear structures in the activation space that can be exploited to circumvent safety alignment. Through detailed visualizations and empirical experiments, they demonstrate that projecting model representations onto lower-dimensional subspaces can mitigate these vulnerabilities while preserving essential information for alignment. The paper proposes two novel fine-tuning methods, FJLT and Bottleneck, to achieve this dimensionality reduction and validate their effectiveness against steering jailbreak attacks.

**Questions To Authors:**

NA

**Reasons To Accept:**

- The paper studies a timely and important problem. Given the increasing attention to the trustworthiness of LLMs, the findings and proposed method could have meaningful impact on future research.
- The paper is well-written and easy to read.
- The paper offers valuable theoretical insights into the relationship between linear structures in the activation space and the model's hidden dimensions.
- Experiments show the effectiveness of the proposed methods.

**Reasons To Reject:**

- Given the scale and complexity of modern LLMs, it remains unclear how well the theoretical analysis based on classical tools such as VC dimension and Rademacher complexity translates to practical settings.
- The proposed method introduces several hyperparameters ($\alpha$ and $K$), yet the paper does not provide a sensitivity analysis or discussion regarding their influence on performance.

---

> ### Author Response · Authors · 2025-06-01
> **Review Response Q1-2**
>
> **Thank you to the reviewer for your insightful review. We address each of your questions below and welcome further discussion.**
>
> **Q1.** Given the scale and complexity of modern LLMs, it remains unclear how well the theoretical analysis based on classical tools such as VC dimension and Rademacher complexity translates to practical settings.
>
> **Response:** We agree that bridging the gap between classical complexity measures and the scale of modern LLMs is an emerging challenge. However, our use of tools such as Rademacher complexity is not intended to yield generalization bounds for billion-parameter models. Rather, we employ these tools to estimate the capacity of simple linear classifiers to identify binary or counterfactual concepts, such as positive vs. negative sentiment, or benign vs. harmful prompts, within the intermediate high-dimensional feature spaces of LLMs, not tied to their overall architecture. This perspective helps us reason about the effectiveness of linear activation steering that results in a model's vulnerability to adversarial manipulation as well as how dimensionality reduction may affect the linear separability of such concepts.
>
> **Q2.** several hyperparameters ($\alpha$ and $K$), yet the paper does not provide a sensitivity analysis or discussion regarding their influence on performance.
>
> **Response:** In our work, we provided a hyperparameter anaysis on the attention heads in the FJLT model and the insertion layers for the Bottleneck model in Appendix C. But we did not include an anlaysis on $\alpha$ and $K$, thank you for pointing that out. In Table 1 and 2 below, we present our ablation study on the hyperameters $\alpha$ and $K$ respectively.
>
> We observe in Table 1 that as $\alpha$ increases, the model's performance increases as well. This suggests that it is important for the Llama2-7B-Chat-Bottleneck model to have sufficient gradients backpropagating from the anchor dataset to be able to distinguish between safe and harmful instructions without a strong linear structure for safety in the lower dimensional projection space. This makes sense since increasing alpha to 1.0 balances the loss equally between the harmful and benign instruction datasets, hence providing the model with enough information to characterize each type of instruction properly.
>
> In Table 2, we find that our models are robust to $K$ since for all values of $K$ tested, the results are relatively similar across all metrics and improve significantly over the baseline. These results are expected, given that the dimension of each projection subspace ($K$) is still substantially smaller than the original hidden dimension of 4096.
>
> Table 1: Hyperparameters, refusal scores, safety scores and perplexity (PPL) on harmful and benign instructions after attacking each Llama2-7B-Chat-Bottleneck model using the ActAdd jailbreak. The direction of increasing or decreasing values signifying better performance is indicated by each metric’s arrow direction.
> |Model|K|$\alpha$|(Harmful inst.) Refusal $\uparrow$|(Harmful inst.) Safety$\uparrow$|(Benign inst.) Refusal$\downarrow$| (Benign inst.) PPL$\downarrow$|
> |--|--|--|:--:|:--:|:--:|:--:|
> |Llama2-7B-Chat-Bottleneck|2048|0.1| $0.36_{\pm{0.02}}$| $0.48_{\pm{0.03}}$| $0.89_{\pm{0.02}}$| $1.88_{\pm{0.26}}$ |
> |Llama2-7B-Chat-Bottleneck|2048|0.5|$0.67_{\pm{0.2}}$ |$0.62_{\pm{0.03}}$ |$0.92_{\pm{0.02}}$ |$1.94_{\pm{0.46}}$ |
> |Llama2-7B-Chat-Bottleneck|2048|1.0| **0.95$_{\pm{0.02}}$**| **0.97$_{\pm{0.01}}$**|**0.30$_{\pm{0.03}}$** | **1.08$_{\pm{0.34}}$**|
>
> Table 2: Hyperparameters, refusal scores, safety scores and perplexity (PPL) on harmful and benign instructions after attacking each Llama2-7B-Chat-Bottleneck model using the ActAdd jailbreak. The direction of increasing or decreasing values signifying better performance is indicated by each metric’s arrow direction.
> |Model|K|$\alpha$|(Harmful inst.)  Refusal $\uparrow$|(Harmful inst.) Safety$\uparrow$|(Benign inst.) Refusal$\downarrow$| (Benign inst.) PPL$\downarrow$|
> |--|--|--|:--:|:--:|:--:|:--:|
> |Llama2-7B-Chat-Bottleneck|1024|1.0|$0.90_{\pm{0.12}}$ |$0.91_{\pm{0.11}}$ |$0.54_{\pm{0.03}}$ |$1.44_{\pm{0.25}}$ |
> |Llama2-7B-Chat-Bottleneck|2048|1.0|$0.95_{\pm{0.02}}$| **0.97$_{\pm{0.01}}$**|**0.30$_{\pm{0.03}}$** | **1.08$_{\pm{0.34}}$**|
> |Llama2-7B-Chat-Bottleneck|3073|1.0|**0.97$_{\pm{0.00}}$** |$0.97_{\pm{0.01}}$ |$0.36_{\pm{0.04}}$ |$1.30_{\pm{0.16}}$ |

---

> > ### Comment · Reviewer_NYCF · 2025-06-05
> >
> > Thank you for your response. The current results show that performance consistently improves as $\alpha$ increases from 0.1 to 1.0. This raises the question of whether further increasing $\alpha$ would continue to improve performance or eventually lead to degradation (according to the results in paper, I would suspect the latter). A proper sensitivity analysis should ideally capture the full trend, including the point where increasing $\alpha$ no longer benefits the model or starts to hurt performance. Without this, the analysis feels incomplete and leaves ambiguity regarding the true impact of this hyperparameter.
> >
> > In addition, I notice that the sensitivity analysis was only conducted on LLaMA2, while the main paper also reports results on Gemma and Qwen2. Given that the optimal results in Table 2 are achieved using different $\alpha$ values across different models, it seems likely that you have already experimented with varying $\alpha$ settings when preparing the submission. Would it be possible to report those results—for example, what are the metrics on Gemma when $\alpha = 1$?
> >
> > I don’t mean to be overly picky as a reviewer, but I personally believe this hyperparameter is quite important for practical use. If these experiments were already conducted during the preparation of Table 2, sharing them should not require significant additional effort and would help provide a more complete understanding of the method’s applicability across different backbones.

---

> > > ### Author Response · Authors · 2025-06-06
> > > **Reply to comment by Reviewer NYCF**
> > >
> > > Thank you for your comments, we agree that $\alpha$ is an important hyperparameter and should be treated carefully. As such, we report our results in Table 1 below when further increasing $\alpha$ to 1.5 on Llama2-7B-Chat-Bottleneck and for $\alpha=1.0$ on Gemma-1.1-7B-IT-Bottleneck.
> > >
> > > As expected, Llama2-7B-Chat-Bottleneck's performance indeed degrades with an increase in $\alpha$ suggesting that $\alpha=1.0$ was the optimal choice. In contrast, Gemma-1.1-7B-IT-Bottleneck performs poorly on harmful instructions with $\alpha=1.0$ since it does not refuse majority of prompts but does well to answer benign instructions. Hence, our final results report using $\alpha=0.1$, to reduce the influence of the benign anchor dataset on the model.
> > >
> > > We appreciate the reviewer's attention to detail regarding our sensitivity analysis as it further improves the quality of our work.
> > >
> > > Table 1: Hyperparameters, refusal scores, safety scores and perplexity (PPL) on harmful and benign instructions after attacking each Llama2-7B-Chat-Bottleneck model and Gemma-1.1-7B-IT-Bottleneck using the ActAdd jailbreak. The direction of increasing or decreasing values signifying better performance is indicated by each metric’s arrow direction.
> > > |Model|K|$\alpha$|(Harmful inst.) Refusal $\uparrow$|(Harmful inst.) Safety$\uparrow$|(Benign inst.) Refusal$\downarrow$| (Benign inst.) PPL$\downarrow$|
> > > |--|--|--|:--:|:--:|:--:|:--:|
> > > |Llama2-7B-Chat-Bottleneck|2048|0.1| $0.36_{\pm{0.02}}$| $0.48_{\pm{0.03}}$| $0.89_{\pm{0.02}}$| $1.88_{\pm{0.26}}$ |
> > > |Llama2-7B-Chat-Bottleneck|2048|0.5|$0.67_{\pm{0.20}}$ |$0.62_{\pm{0.03}}$ |$0.92_{\pm{0.02}}$ |$1.94_{\pm{0.46}}$ |
> > > |Llama2-7B-Chat-Bottleneck|2048|1.0| **0.95$_{\pm{0.02}}$**| **0.97$_{\pm{0.01}}$**|**0.30$_{\pm{0.03}}$** | **1.08$_{\pm{0.34}}$**|
> > > |Llama2-7B-Chat-Bottleneck|2048|1.5|0.81$_{\pm{0.03}}$|0.81$_{\pm{0.03}}$|0.60$_{\pm{0.03}}$|1.08$_{\pm{0.10}}$|
> > > |Gemma-1.1-7B-IT-Bottleneck|2048|1.0|0.01$_{\pm{0.00}}$|0.39$_{\pm{0.08}}$|0.48$_{\pm{0.02}}$|2.52$_{\pm{0.24}}$|

---

> > > > ### Comment · Reviewer_NYCF · 2025-06-10
> > > >
> > > > Thank you for your response. Based on the additional results you provided, I personally feel that the hyperparameter $\alpha$ is quite sensitive and requires very careful tuning in practice. This is not a good sign, as it may weaken the practicality of your method.

---

> > > > > ### Author Response · Authors · 2025-06-10
> > > > >
> > > > > We appreciate your feedback and understand the concern regarding the hyperparameter $\alpha$. In our experiments, we explored $\alpha$ values of 0.1, 0.5, 1.0, and 1.5. Notably, all of these choices yielded substantial performance improvements over the baseline.
> > > > >
> > > > > While $\alpha$ does influence the final performance, we found the method to be reasonably robust within this range. Moreover, tuning $\alpha$ is relatively inexpensive: for most models, satisfactory results can be achieved with fewer than 30 epochs of fine-tuning, and in some cases (including models up to 7B parameters), as few as 5 epochs are sufficient to identify a good setting.
> > > > >
> > > > > We believe this level of tuning is comparable to (or even less demanding than) common practices in fine-tuning large models, and does not substantially diminish the practicality of our method.

---

> > ### Author Response · Authors · 2025-06-11
> >
> > Further following up on this point, we provide an estimated total run time of each model for each fine-tuning run in Table 1 and results using a more fine-grained tuning of $\alpha$ in Table 2. In Table 1, we see that each run takes 4-30 mins and tuning $\alpha$ carefully over multiple runs will not be too computationally costly. In Table 2, we observe that over smaller increments (0.1) of $\alpha$, our model is robust to the hyperparameter and does not require significant tuning. While tuning over the range of 0.1,0.5,1.0, and 1.5 does seem to have an impact on our method, since $\alpha$ is insensitive to smaller incremements, 0.1 instead of 0.5, we are not required to tune $\alpha$ that carefully. This reduces the number of values needed to test for the optimal $\alpha$ and does not take a significant amount of resources as seen in Table 2.
> >
> > Table 1: Estimated total run time per fine-tuning run for each model and different number of H100 GPUs. Generally, Bottleneck models will take a longer time due to the additional anchor dataset.
> > |Model|Epochs|Total run time (min)|Number of GPUs|
> > |-|-|-|-|
> > |Llama2-7B-Chat-FJLT|25|5| 4|
> > |Llama2-7B-Chat-Bottleneck|30|30| 4|
> > |Gemma-1.1-7B-IT-FJLT|5|4|4|
> > |Gemma-1.1-7B-IT-Bottleneck|5|10|4|
> > |Qwen2-7B-Instruct-FJLT| 5 |4 | 4
> > |Qwen2-7B-Instruct-Bottleneck| 20 | 15 | 4
> >
> > Table 2: Hyperparameters, refusal scores, safety scores and perplexity (PPL) on harmful and benign instructions after attacking each Llama2-7B-Chat-Bottleneck model using the ActAdd jailbreak. The direction of increasing or decreasing values signifying better performance is indicated by each metric’s arrow direction.
> > |Model|K|$\alpha$|(Harmful inst.) Refusal $\uparrow$|(Harmful inst.) Safety$\uparrow$|(Benign inst.) Refusal$\downarrow$| (Benign inst.) PPL$\downarrow$|
> > |--|--|--|:--:|:--:|:--:|:--:|
> > |Llama2-7B-Chat-Bottleneck|2048|0.8| **0.97$_{\pm{0.01}}$**| 0.97$_{\pm{0.01}}$|0.31$_{\pm{0.01}}$|1.27$_{\pm{0.28}}$|
> > |Llama2-7B-Chat-Bottleneck|2048|0.9|0.95$_{\pm{0.03}}$|**0.99$_{\pm{0.03}}$**|**0.30$_{\pm{0.01}}$**|1.12$_{\pm{0.36}}$|
> > |Llama2-7B-Chat-Bottleneck|2048|1.0| 0.95$_{\pm{0.02}}$| 0.97$_{\pm{0.01}}$|0.30$_{\pm{0.03}}$ | **1.08$_{\pm{0.34}}$**|
> > |Llama2-7B-Chat-Bottleneck|2048|1.1|0.95$_{\pm{0.12}}$|0.96$_{\pm{0.02}}$|0.33$_{\pm{0.01}}$|1.09$_{\pm{0.30}}$|
> > |Llama2-7B-Chat-Bottleneck|2048|1.2|**0.97$_{\pm{0.12}}$**|0.98$_{\pm{0.02}}$|**0.28$_{\pm{0.01}}$**|1.09$_{\pm{0.30}}$|

---

### Official Review · Reviewer_ii6K · 2025-05-13

**Rating:** 7
**Confidence:** 3
**Ethics Flag:** 1

**Summary:**

This paper seeks to mitigate an attack (AddAct; Arditi et al., 2024) that attempts to erase the concept vector associated with "safety"; the mitigation involves projecting the activation space to a lower dimensional one, such that the new subspace is not linearly separable. Evaluations over three instruction-tuned LLMs (Llama 2, Gemma 1.1, Qwen2) on JailbreakBench demonstrate the efficacy of the mitigation on harmful instructions while refusing fewer benign instructions than standard finetuning.

**Questions To Authors:**

Please address any questions above / feel free to correct any misunderstandings. Thanks!

**Reasons To Accept:**

1. Activation steering to jailbreak LLM safety refusals is an active area of research; these methods are largely reliant on linear separability of a concept (e.g., "safety" or "refusal") in the embedding space. As such, parallel research on mitigations is highly relevant; the approach here is simple but logical and demonstrates a solid proof of concept. (As far as I'm aware, the contribution is novel, though I'll defer to others on this point; likewise with appropriate baseline comparisons.)

2. The experiments in section 5, supplemented by the appendices, are reasonably thorough; while one could always ask for more, these suffice to convey the important result: the proposed mitigation significantly improve the refusal rate & safe generations (per LlamaGuard) on harmful prompts while refusing fewer benign prompts in comparison to finetuning on the jailbreak dataset.

3. The paper is a pleasant read -- generally clear and logically organized.

**Reasons To Reject:**

1. The paper is a bit short on details when it comes to the visualized concepts in Fig. 2 (emotion) and Fig. 4 (truthfulness, emotion, and safety) -- where do the data points and labels come from?

---

> ### Author Response · Authors · 2025-06-01
> **Review Response Q1**
>
> **Thank you to the reviewer for your insightful review. We address your question below and welcome further discussion.**
>
> **Q1.** The paper is a bit short on details when it comes to the visualized concepts in Fig. 2 (emotion) and Fig. 4 (truthfulness, emotion, and safety) -- where do the data points and labels come from?
>
> **Response:** Thank you for pointing that out, we should have included these details in the appendix and will add it in the revision. For each of the visualized concepts, we list where the datapoints and labels originate from below:
>
> 1) **Truthfulness:** We used the cities and neg_cities datasets from [1]. Cities contain true statements of world cities belonging to the countries they are in. For example, "The city of Sevastopol is in Ukraine." While neg_cities are false statements of the same template regarding cities and countries. The labels follow the truth (blue) and falsity (red) of each statement.
> 2) **Emotion:** We used the anger, disgust, happiness and surprise datasets from [2]. These datasets contain sentences representing each emotion and we consider anger and disgust to be negative (red), while happiness and surprise are positive (blue).
> 3) **Safety:** We used the harmful (red) instructions from JailbreakBench [3], AdvBench [4] and HarmBench [5] and safe (blue) instructions from Alpaca [6].
>
> [1]: Marks and Tegmark (2024): https://arxiv.org/abs/2310.06824, https://github.com/saprmarks/geometry-of-truth \
> [2]: Zou et. al. (2023): https://arxiv.org/abs/2310.01405, https://github.com/andyzoujm/representation-engineering/tree/main/data/emotions \
> [3]: Chao et al. (2024): https://arxiv.org/abs/2404.01318, https://github.com/jailbreakbench/jailbreakbench \
> [4]: Zou et al. (2023b):https://arxiv.org/abs/2307.15043, https://github.com/llm-attacks/llm-attacks \
> [5]: Mazeika et al. (2024): https://arxiv.org/abs/2402.04249, https://github.com/centerforaisafety/HarmBench \
> [6]: Taori et al. (2023): https://github.com/tatsu-lab/stanford_alpaca

---

> > ### Comment · Reviewer_ii6K · 2025-06-09
> >
> > A belated thank you for the clarification on visualized concepts -- would love to see this in the paper/appendix if possible. (Perhaps out of scope for this paper: it could be interesting to understand how behavior might differ for examples of aleatoric uncertainty -- e.g., for geographical truth, are representations of disputed territories more likely to be compromised? likewise for examples of safe/unsafe text that are context-dependent, etc.)

---

> > > ### Author Response · Authors · 2025-06-09
> > >
> > > Thank you for your endorsement and reply! We will include the dataset details in the appendix in the final revision. Your point on aleatoric uncertainty is well-taken — unlike safety cases with relatively clear labels, context-dependent or disputed examples (e.g., geopolitical facts) may exhibit more nuanced behavior and warrant special attention. A promising direction for future work.

---

### Official Review · Reviewer_bcLY · 2025-05-14

**Rating:** 7
**Confidence:** 4
**Ethics Flag:** 1

**Summary:**

This submission studies how the increasing dimensionality of LMs leads to emergent linear representations that could be exploited to circumvent safety alignment.

The authors demonstrate that larger models exhibit more linearly separable safety-relevant concepts, enabling attacks from piror work. They propose two defenses that reduce jailbreak success while preserving model performance.

The work is interesting, with solid empirical support and theoretical grounding, though the motivation and assumptions around linearity could be more clearly articulated and weaken the paper in its current form.

**Questions To Authors:**

Depending on the rebuttal and what uncertainties are addressed, I will increase my score. My main concerns are related to the motivation, how well this enables future safety work/model deployment, and the relation to prior work that potentially does similar things.

* Your work assumes that safety-relevant concepts are linearly encoded in large language models and focuses on defending against jailbreak attacks that exploit this linearity. How do your methods generalize to scenarios where safety-relevant features are encoded non-linearly or in a distributed fashion, as explored in, for example, https://arxiv.org/abs/2405.14860? Have you evaluated whether your defenses preserve performance in cases where linear representations may not be dominant?

* Following up on that, where do you expect the usability limit to lie between linear and non-linearly encoded representations? And what happens when multiple linearly encoded directions must be simultaneously used due to context-dependent complexity?

* The motivation and framing of the “paradox of linear separability” are unclear. Is the threat model primarily about jailbreakability increasing in open-source models as they scale, because of easier fine-tuning and activation-level manipulation? If so, the argument seems to underweight how these same linear structures might enable more effective safety mechanisms. Could you clarify your intended argument and how this informs model development or deployment safety policy?

* Following up on the prior question, to what extend would making LMs vastly deeper with the same number of parameters affect safety robustness? (Or more general, what is the impact of your work on understanding representation robustness across layers?)

* This paper shares conceptual similarities with https://arxiv.org/abs/2302.12461, which also exploit linear activation structures to analyze and manipulate model behavior in the context of backdoors. Both works use dimensionality reduction to disrupt harmful behaviors. Could your method support similar localized interpretability or editing, and how might it complement approaches as in https://arxiv.org/abs/2302.12461?

* Why is your dimensionality reduction applied only to attention outputs and not MLP outputs, especially given prior work showing MLPs often encode key semantic features? Would projecting from MLPs offer similar or stronger protection?

* How well do the proposed defenses generalize to more complex, context-dependent, or non-linearly encoded safety-relevant features that go beyond such simplified scenarios (positive/negative)?

**Reasons To Accept:**

* Offers a well-supported analysis of how high-dimensional representations contribute to jailbreak vulnerabilities in LMs. I also appreciate that they tested multiple open-source models from different families.

* Proposes simple, effective defenses that reduce jailbreak success without significant utility loss.

* Combines empirical evidence and theoretical justification to support its central claims.

**Reasons To Reject:**

* Assumes linearity of safety-relevant representations without addressing non-linear or distributed alternatives.

* The motivation and framing of the core threat model (paradox of linear separability) are seemingly underdeveloped.

* Defense mechanisms are narrowly evaluated and may not generalize to more complex or non-linear threat settings. (Evaluation primarily focuses on binary concepts like “positive” and “negative” sentiment, which are relatively easy to linearly separate)

---

> ### Author Response · Authors · 2025-06-01
> **Review Response Q1**
>
> **Thank you to the reviewer for your detailed review. We address each of your questions below and welcome further discussion.**
>
> **Q1.** Your work assumes that safety-relevant concepts are linearly encoded in large language models and focuses on defending against jailbreak attacks that exploit this linearity. How do your methods generalize to scenarios where safety-relevant features are encoded non-linearly or in a distributed fashion, as explored in, for example, https://arxiv.org/abs/2405.14860? Have you evaluated whether your defenses preserve performance in cases where linear representations may not be dominant?
>
> **Response (1/2):** From our empirical evidence (Figures. 4,5,6,7), we find that one of the simplest safety-relevant concepts, a binary decision in determining if a prompt is harmful or safe, is linearly encoded in popular LLMs such as Llama2-7B-Chat, Gemma-1.1-7B-IT and Qwen2-7B-Instruct. However, we agree that there are likely to be more nuanced safety-related concepts that are non-linear and there are many jailbreak methods that do not simply exploit a strong linear structure [1,2,3]. We also note that there is a subtle distinction in that these jailbreak methods do not explicitly target the non-linearity of those safety-related concepts either, but may be implicitly using the non-linear features. From this perspective, we decide to test our method against the Greedy Coordinate Gradient (GCG) attack [4,5] and report our results below in Table 1. We report the Attack Success Rate (ASR) of GCG which is the percentage of harmful prompts that each model did not refuse, a lower percentage is better.
>
> Table 1: Attack Success Rate (ASR) of the Greedy Coordinate Gradient (GCG) attack on the baseline Llama-2-7B-Chat model and models with our methods (FJLT and Bottleneck) implemented. A lower percentage is better.
> |Model|ASR (%) $\downarrow$|
> |--|--|
> |Llama-2-7B-Chat (baseline)|42|
> |Llama-2-7B-Chat-FJLT|53|
> |Llama-2-7B-Chat-Bottleneck|43|
>
> We observe that the Bottleneck method does not hurt nor help the model defend against GCG while the FJLT method hurts the model to a small degree. It is most likely because while GCG may be implicitly using the non-linear features for jailbreaking, the attack does not explicitly use their structure or geometry, and our projection did not disrupt or remove these non-linear features. This could be considered a limitation of our generalization ability to attacks that do not rely on linear features since it is unclear if there is a relationship between dimension and those types of attacks. The generalization of our method is also limited by the data that is available for fine-tuning and the targeted features. We require that the data used during fine-tuning be general enough to preserve utility while strongly capturing the necessary features to be retained. For example, when fine-tuning our Bottleneck models, we need to create anchor datasets for each of the aligned models we were going to fine-tune since the anchor needed to contain sufficient benign responses generated by the same model family.

---

> > ### Author Response · Authors · 2025-06-01
> > **Review Response Q1**
> >
> > **Response (2/2):** We further tested our method on Qwen2-0.5B-Instruct that has weaker linear structures as seen in Figures. 2, 3 and in the anonymous link, [qwen2-0.5b-instruct](https://anonymous.4open.science/r/colm2025-37DB/qwen2-0.5b-instruct.pdf) and report our results below in Table 2. Our link contains an image of PCA plots with hidden representations of safe (blue) and harmful (red) prompts from Llama2-7B-Chat (left) and Qwen2-0.5B-Instruct (right). It is clear that there is weaker linear separation in the smaller Qwen2-0.5B-Instruct as compared to the Llama2-7B-Chat model. From our results, we observe non-trivial performance gains by our models across all metrics, despite the weak linearity of safety-relevant features. In particular, refusal and safety scores in our Qwen2-0.5B-Instruct-FJLT model improved over the baseline by about 50% on harmful instructions.
> >
> > Table 2: Refusal scores, safety scores and perplexity (PPL) on harmful and benign instructions after attacking each model using the ActAdd jailbreak. We also include results on the baseline Instruct model without jailbreaks for reference. The FT model considered is the baseline model fine-tuned using FJLT experimental settings. The direction of increasing or decreasing values signifying better performance is indicated by each metric’s arrow direction.
> > |Model + ActAdd|(Harmful inst.) Refusal $\uparrow$|(Harmful inst.) Safety$\uparrow$|(Benign inst.) Refusal$\downarrow$| (Benign inst.) PPL$\downarrow$|
> > |--|:--:|:--:|:--:|:--:|
> > |Qwen2-0.5B-Instruct (baseline)| 0.01$_{\pm 0.00}$ | 0.27$_{\pm 0.00}$ | 0.91$_{\pm 0.00}$ | 1.41$_{\pm 0.00}$
> > |Qwen2-0.5B-Instruct-FT | 0.19$_{\pm 0.01}$ | 0.50$_{\pm 0.02}$ | 0.99$_{\pm 0.01}$ | **1.29$_{\pm 0.01}$**
> > |Qwen2-0.5B-Instruct-FJLT (K=96) | **0.54$_{\pm 0.22}$** | **0.71$_{\pm 0.18}$** | 0.93$_{\pm 0.04}$ | 1.39$_{\pm 0.02}$
> > |Qwen2-0.5B-Instruct-Bottleneck| 0.11$_{\pm 0.07}$ | 0.32$_{\pm 0.03}$ | **0.67$_{\pm 0.04}$** | 1.37$_{\pm 0.04}$|
> >
> > [1]: Yi et. al. (2024): https://arxiv.org/abs/2407.04295 \
> > [2]: Zhou et. al. (2025): https://arxiv.org/abs/2502.11379 \
> > [3]: Wei et. al. (2024): https://arxiv.org/abs/2310.06387 \
> > [4]: Zou et. al. (2023): https://arxiv.org/abs/2307.15043 \
> > [5]: Zhou et. al. (2024): https://arxiv.org/abs/2403.12171, http://easyjailbreak.org/

---

> ### Author Response · Authors · 2025-06-01
> **Review Response Q2-4**
>
> **Q2.** Following up on that, where do you expect the usability limit to lie between linear and non-linearly encoded representations? And what happens when multiple linearly encoded directions must be simultaneously used due to context-dependent complexity?
>
> **Response:** We expect that it depends on how the structure of the non-linearly encoded representations are used. If a jailbreak or steering method relies heavily on the structure of the representations, then projecting to a low dimensional space will most likely break or at least distort the non-linear structure which could result in a decent defence against steering.
>
> For example, in https://arxiv.org/abs/2405.14860, where circular representations of days/months are explored and intervened on, if projecting the model's feature vectors into a lower dimensional subspace disrupts that circular structure, it is likely that the activation patching used in the paper would be less effective.
>
> In the context of multiple linearly-encoded directions that must be used simultaneously, we do expect the utility of our method to be quite high since simply removing the one of their linear structures would suffice and that is clearly achievable from our experiments.
>
> **Q3.** The motivation and framing of the “paradox of linear separability” are unclear. Is the threat model primarily about jailbreakability increasing in open-source models as they scale, because of easier fine-tuning and activation-level manipulation? If so, the argument seems to underweight how these same linear structures might enable more effective safety mechanisms. Could you clarify your intended argument and how this informs model development or deployment safety policy?
>
> **Response:** Indeed the threat model is as you say, primarily about the increasing ease of jailbreak in open-source models as they scale, due to activation-level manipulation and we agree that these linear structures can be used positively as in representation engineering [1]. However, we argue that having the freedom to do so does not outweigh the potential harm that can come from exploiting them with ill intent. For example, if a user interacts with a jailbroken llm that generates outputs endorsing unethical or harmful perspectives, it can influence the user to engage in harmful behaviors that they might not have otherwise [2,3]. We refer to Appendix D.4 in our submission as an illustration of persuasive generated text promoting harmful actions.
>
> Our motivation is that as open-source models scale, it becomes easier to perform activation steering. If used with good intent, it can be a benefit. But is it possible to prevent a threat from using it maliciously? It would seem not, unless we remove those linear structures altogether, thereby also preventing effective safety mechanisms that may arise from those linear structures. Yet, it is a worthy trade-off in open-sourced models as it prevents users from exploiting the linearity for more harm than good.
>
> However, we should not remove all linear structures in the model, since they can be an advantage, used to alter a model's behavior [4]. Accordingly, our method aims to only remove the linear structure pertaining to safety-relevant features, hence providing an additional layer of deterrence in ensuring a model's safety policy.
>
> [1]: Zou et. al. (2023): https://arxiv.org/abs/2310.01405 \
> [2]: Weidinger et. al. (2021): https://arxiv.org/abs/2112.04359 \
> [3]: Breum et. al. (2023): https://arxiv.org/abs/2312.15523 \
> [4]: Rimsky et. al. (2024): https://aclanthology.org/2024.acl-long.828/
>
> **Q4.** Following up on the prior question, to what extend would making LMs vastly deeper with the same number of parameters affect safety robustness? (Or more general, what is the impact of your work on understanding representation robustness across layers?)
>
> **Response:** Our work mainly targets a model's hidden dimension and we do not observe a clear or intuitive relationship between layer depth and linear structures. However, maintaining the same number of parameters while vastly increasing depth should result in a much lower dimensional activation space that our work shows to have a weaker linear structure (Figures. 2,3). The upshot of that is an improved defence against any negative exploitations that rely on manipulating the activation vectors (not limited to jailbreaking and safety). Hence, there is an increased representation robustness across the layers.
>
> Furthermore, the models in our experiments range from 24 to 32 layers, illustrating the effectiveness of our method in improving robustness across varying model depths.

---

> ### Author Response · Authors · 2025-06-01
> **Review Response Q5-7**
>
> **Q5.** This paper shares conceptual similarities with https://arxiv.org/abs/2302.12461, which also exploit linear activation structures to analyze and manipulate model behavior in the context of backdoors. Both works use dimensionality reduction to disrupt harmful behaviors. Could your method support similar localized interpretability or editing, and how might it complement approaches as in https://arxiv.org/abs/2302.12461?
>
> **Response:** Thank you for sharing this relavent work with us. It could be possible to interpret our method as localizing the safety linear structures to the components in the model where we implement our modification to prevent their manipulation. For example, in the FJLT model, we only insert the FJLT modification in one attention head in each layer and this is chosen through experimentation. We can interpret the chosen head as the local component in the model that has significant contribution to the model's overall safety alignment since modifying this component resulted in an improved jailbreak defence. Similarly, we can say the same for the layer chosen for insertion of the linear autoencoder in the Bottleneck models or those token positions/layer activations where we observe the safety linear structure in our empirical experiments.
>
> Our work is complementary to their work in 2 aspects:
> 1) **The fundamental approach to dimensionality reduction:** Our work is largely inspired by the Fast Johnson–Lindenstrauss Transform (FJLT) and lemma that allows us to approximately preseve the outputs in the attention module with a smaller activation space, as well as simply learning an effective lower dimensional subspace to project into through a linear autoencoder (Bottleneck). While [Lamparth and Reuel (2024)](https://arxiv.org/abs/2302.12461) uses a low rank Principle Component Analysis (PCA) subspace for their projection.
> 2) **Maintaining model utility:** The distinction made in the previous point leads us to our second point. Since a low rank PCA subspace is used in [Lamparth and Reuel (2024)](https://arxiv.org/abs/2302.12461), there is significant information loss that leads to an increase in validation loss and reduced performance. On the other hand, in our work, while we do observe some utility loss on specialized tasks in our FJLT model, there is minimal loss on general instruction datasets. Further, we maintain most utility across all tasks in the Bottleneck models. We prioritised minimizing such losses to ensure that even with our dimensionality reductions, our models remain practical and effective.
>
> **Q6.** Why is your dimensionality reduction applied only to attention outputs and not MLP outputs, especially given prior work showing MLPs often encode key semantic features? Would projecting from MLPs offer similar or stronger protection?
>
> **Response:** In our FJLT models, we insert the FJLT projection between the Query and Key matrices in an attention head as we are motivated by the Johnson-Lindenstrauss (JL) lemma to capitalize on the preservation of the dot-product and hence, the attention matrix, in a lower dimensional space. In our Bottleneck models, we insert the linear autoencoder after some layer in the model and the layer that we consider here is a decoder block that consists of both the attention and MLP layers. Therefore, we do actually project from the MLPs. From our experiments, we observe improvements when projecting from both the attention layer and the MLP layer (Table 1,2) and indeed better results when projecting from the MLPs (Bottleneck model). We apologize for the lack of clarity in our text and will improve that in the next iteration.
>
> **Q7.** How well do the proposed defenses generalize to more complex, context-dependent, or non-linearly encoded safety-relevant features that go beyond such simplified scenarios (positive/negative)?
>
> **Response:** We observe from Q1, Table 2 that our defences generalize relatively well in the Qwen2-0.5B-Instruct model, where the encoded safety features only have a weak linear structure. Despite that, we were still able to improve its defence notably against the steering jailbreak.
> Regarding the Greedy Coordinate Gradient (GCG) attack, Q1, Table 1, where there is no exploitation of the linear structure of safety-relevant features, our method has almost no effect. This could be considered a limitation of our proposed defences and we may not generalize well to such attacks.
> However, in the cases where jailbreak methods rely heavily on either multiple linearly encoded safety-relevant features or the structure of non-linear ones, we have assuring empirical evidence that projecting the representation space into a smaller subspace will disrupt the geometry enough to alleviate such attacks.

---

> ### Author Response · Authors · 2025-06-07
> **Any further questions from Reviewer bcLY?**
>
> We would like to thank the reviewer again for your thoughtful review and valuable feedback.
>
> We would appreciate it if you could let us know if our responses have addressed your concerns and whether you still have any other questions about our rebuttal.
>
> We would be happy to do any follow-up discussion or address any additional comments.

---

> ### Comment · Reviewer_bcLY · 2025-06-10
> **Rebuttal Acknowledgement and Score Increase**
>
> Thank you for your thorough replies. In short, I've increased my score to 7 and raised my confidence, based on an advancement of trust, as I believe this submission is thorough (post-rebuttal) and raises an important debate (backed up by their findings).
>
> For the final version, I would ask the authors to include the following (besides their new results):
>
> * Clarifying statements related to my questions, especially regarding the motivation and threat model
>
> * I think this paper has the potential to guide some of the interpretability research projects conducted by junior researchers and can help summarize past approaches and limitations of linear separability based on their findings. This point is essentially why I mentioned the backdoor paper that I felt did something similar in concept, although simpler. So I think a discussion (in the appendix) on what their findings mean for similar works on linear separability and robstuness (e.g., like the backdoor paper but ideally add more works), how FJLT solves some of these issues, and how non-linear features + context-dependent complexity with multiple directions remain an issue could be very helpful for the field.

---

> > ### Author Response · Authors · 2025-06-11
> >
> > Thank you for your endorsement! We appreciate your encouragement and will include our new results, the clarification on our motivation and threat model, and additional discussions on similar works on linear separability and robustness, as well as the remaining open issues, in our revision.

---

### Author Response · Authors · 2025-06-11
**Global Response: Summary of Our Discussion with Reviewers (1/2)**

We sincerely thank the reviewers for their thoughtful feedback and constructive suggestions, helping to improve the quality of our work. During the rebuttal period, we made several clarifications and additions to strengthen our submission. Below, we summarize the key points addressed:

1. **Clarifying the Role of Classical Tools** \
   In response to Reviewer NYCF’s question on the applicability of classical complexity measures like Rademacher complexity, we clarified that these tools were not used to derive generalization bounds for large LLMs. Instead, we used them to reason about the separability of safety-relevant concepts in high-dimensional internal representations. This analysis supports our understanding of activation steering and the effects of dimensionality reduction in hindering adversarial exploits.
2. **New Hyperparameter Ablation Studies** \
   We extended our experimental evaluation by conducting new ablation studies on two key hyperparameters, \$\alpha\$ and \$K\$, previously underexplored in the main text. The results (included in new Tables 1 and 2) reveal how increasing \$\alpha\$ improves safety outcomes by balancing training loss between harmful and benign instructions.

    While we acknowledge that our method introduces a new hyperparameter $\alpha$, we stress that this tuning burden is comparable to -- or lighter than -- widely accepted practices in LLM fine-tuning. For example, learning rate, dropout, weight decay, and decoding temperature are all known to significantly affect performance and require grid search or multiple runs. In contrast, our ablation shows that $\alpha$ is robust, outperforming the baseline, across a reasonable range and can be tuned efficiently (within 5–30 epochs of lightweight fine-tuning). This suggests that the added overhead is not a significant obstacle in practice.

3. **Positioning Jailbreaking Within a Broader Framework** \
   Addressing Reviewer 6omJ’s concerns, we clarified that our focus on jailbreaking serves as a compelling proof-of-concept for a broader research agenda: mitigating linear exploitability of activation directions in LLMs. Jailbreaking is one high-impact manifestation of this phenomenon. While prior works aim to erase certain concepts (e.g., gender) from internal representations, our approach instead entangles safety-relevant concepts in lower-dimensional spaces, preserving the necessary safety information while impeding adversarial access via linear steering.

4. **Scope Beyond Jailbreaking: Hallucinations and Calibration** \
   We acknowledge the reviewer’s insight that other failure modes--such as hallucination and calibration--warrant similar scrutiny. We surveyed relevant literature and discussed why current theoretical understandings of hallucination (e.g., stemming from data rarity or temporal drift) are not easily mapped to representational dimensionality. While our framework does not yet extend to hallucination mitigation, we agree that exploring this direction through the lens of representational geometry is a promising future avenue.

5. **Extending Beyond Linearity** \
    In regard to the concerns raised by Reviewer bcLY, we discuss the generalization of our method beyond simple, linear concepts. In our work, we demonstrate the effectiveness of our method in defending against attacks that exploit the linearity of safety-relevant concepts, even if these are only weakly present. However, in scenarios where there are complex, context-dependent, or non-linearly encoded safety-relevant features, our method may not generalize well. Primarily because it is unclear if there is a relationship between dimension and those types of attacks, since they do not explicitly rely on the linear structure of concepts. This would be an interesting direction for future work.

    However, in the cases where jailbreak methods rely heavily on either multiple linearly encoded safety-relevant features or the structure of non-linear ones, we have assuring empirical evidence that projecting the representation space into a smaller subspace will disrupt the geometry enough to alleviate such attacks.

6. **Motivation of Our Work** \
    Further, we clarify the motivation of our work. As open-source models scale, it becomes easier to perform activation steering which correlates with the increased dimension of the representation space. While beneficial uses of activation steering exist, the risk of misuse by increasing the ease of jailbreak due to activation-level manipulation outweighs the benefits. However, it does not seem possible to prevent a threat from using it maliciously, unless we remove the linear structures pertaining to safety-relevant features (not all linear structures) altogether. Thereby, also preventing potential safety mechanisms that may arise from safety-related linear structures. Yet, we consider this a worthy trade-off, in providing an additional layer of deterrence to ensure a model's safety policy.

---

> ### Author Response · Authors · 2025-06-11
> **Global Response: Summary of Our Discussion with Reviewers (2/2)**
>
> 7. **Providing Additional Experimental Details** \
>     Reviewer ii6K pointed out that our submission was missing key details regarding the visualized concepts in Figures. 2 and 4. We provided additional clarifications on the data points and labels used in these figures and will include them in our revised manuscript.
>
> In summary, our rebuttal period was dedicated not only to addressing specific reviewer concerns but also to expanding the empirical and conceptual scope of our submission. We hope these efforts clarify our contributions and the broader vision underlying our work.

---

### Decision · Program_Chairs · 2025-07-08

**Decision:**

Accept

**Comment:**

The paper brings up an important problem which is that as LLMs grow in scale, they become more susceptible to linear steering attacks due to their parameter redundancy that translates to easier linear separability for various concepts. Focusing on safety alignment, the paper shows that larger models are indeed more sensitive to such attacks. It then proposes methods to alleviate such issues by fine tuning a model to use lower dimensional activations.

Reasons to accept:
- The paper is thorough and includes substantial experiments and results across various models and datasets.
- The claims in the paper are sound and timely.
- The paper is written clearly and cites relevant work.
- The reviewers are in consensus that the paper is publication-worthy.

Reasons to reject:
- The suggested methods do seem to also harm the quality of the model on some evals; a more thorough analysis of the errors would be interesting.